# Vitamin D metabolites and the gut microbiome in older men

Robert L. Thomas[1,15], Lingjing Jiang[2,15], John S. Adams [3], Zhenjiang Zech Xu[4], Jian Shen[2], Stefan Janssen [5], Gail Ackermann[6], Dirk Vanderschueren[7,8], Steven Pauwels[8,9,10], Rob Knight [6,11,12,13], Eric S. Orwoll [14] & Deborah M. Kado [1,2✉]

The vitamin D receptor is highly expressed in the gastrointestinal tract where it transacts gene expression. With current limited understanding of the interactions between the gut microbiome and vitamin D, we conduct a cross-sectional analysis of 567 older men quantifying serum vitamin D metabolites using LC-MSMS and defining stool sub-Operational Taxonomic Units from16S ribosomal RNA gene sequencing data. Faith's Phylogenetic Diversity and non-redundant covariate analyses reveal that the serum $1,25(OH)_2D$ level explains 5% of variance in α-diversity. In β-diversity analyses using unweighted UniFrac, $1,25(OH)_2D$ is the strongest factor assessed, explaining 2% of variance. Random forest analyses identify 12 taxa, 11 in the phylum Firmicutes, eight of which are positively associated with either $1,25(OH)_2D$ and/or the hormone-to-prohormone $[1,25(OH)_2D/25(OH)D]$ "activation ratio." Men with higher levels of $1,25(OH)_2D$ and higher activation ratios, but not $25(OH)D$ itself, are more likely to possess butyrate producing bacteria that are associated with better gut microbial health.

[1] Department of Medicine, University of California San Diego, La Jolla, CA, USA. [2] Herbert Wertheim School of Public Health and Human Longevity Science, University of California San Diego, La Jolla, CA, USA. [3] Departments of Orthopaedic Surgery and Molecular, Cell and Developmental Biology at UCLA, Los Angeles, CA, USA. [4] State Key Laboratory of Food Science and Technology, Nanchang University, Nanchang, China. [5] Algorithmic Bioinformatics, Department of Biology and Chemistry, Justus-Liebig-University, Gießen, Germany. [6] Department of Pediatrics, University of California San Diego, La Jolla, CA, USA. [7] Department of Chronic Diseases, Metabolism and Ageing (CHROMETA), Laboratory of Clinical and Experimental Endocrinology, KU Leuven, Herestraat 49, B-3000 Leuven, Belgium. [8] Department of Laboratory Medicine, University Hospitals Leuven, Herestraat 49, B-3000 Leuven, Belgium. [9] Department of Cardiovascular Sciences, KU Leuven, Leuven, Belgium. [10] Department of Laboratory Medicine, Jessa Hospital, Hasselt, Belgium. [11] UC San Diego Center for Microbiome Innovation, La Jolla, CA, USA. [12] Department of Bioengineering, University of California San Diego, La Jolla, CA, USA. [13] Department of Computer Science and Engineering, University of California San Diego, La Jolla, CA, USA. [14] Department of Medicine, Bone and Mineral Unit, Oregon Health & Sciences University, Portland, OR, USA. [15] These authors contributed equally: Robert L. Thomas, Lingjing Jiang. ✉email: dkado@ucsd.edu

Several studies suggest that gut microbiota alter intestinal vitamin D metabolism (VDM), and probiotic supplements can affect circulating vitamin D levels[1–6]. These findings are of major clinical interest because multiple large epidemiological studies have shown that persons with low serum vitamin D levels are at increased risk of multiple adverse health outcomes including osteoporosis, obesity, inflammatory bowel disease, incident diabetes, cardiovascular disease, cancer, and autoimmune diseases[1,7]. While some studies have reported associations between low 25-hydroxyvitamin D (25(OH)D: the sum of 25-hydroxyvitamin $D_3$ and 25-hydroxyvitamin $D_2$) levels and disease, others such as those previously conducted in the MrOS Study have revealed no significant association with conditions such as cardiovascular disease or incident diabetes[8–10]. A recent large randomized controlled trial of vitamin D supplementation of over 25,000 adults demonstrated no benefit in preventing cardiovascular events or cancer[11,12].

Because the serum 25(OH)D correlates with overall vitamin D storage, it is the preferred clinical measure to assess vitamin D sufficiency. Clinically, serum 25(OH)D levels ≥20 ng/ml are considered adequate while 25(OH)D levels <20 ng/ml are defined as vitamin D deficiency. However, it is the active form of vitamin D, $1,25(OH)_2D$, that interacts specifically with the vitamin D receptor (VDR) and transacts gene expression. Under tight feedback control, $1,25(OH)_2D$ also induces the expression of catabolic 24-hydroxylase that converts 25(OH)D to $24,25(OH)_2D$ and $1,25(OH)_2D$ to $1,24,25(OH_2D)$ metabolites[13]. Ratios of vitamin D activation ($1,25(OH)_2D/25(OH)D$) and catabolism ($24,25(OH)_2D/25(OH)D$) quantify the proportion of vitamin D stores that are being processed (activated or catabolized) and can serve as a measure of vitamin D mobilization for use in endocrine signaling, or vitamin D flux. These ratios relative to total vitamin D stores based upon 25(OH)D levels may be better predictors of clinically important outcomes including incident hip fracture and earlier mortality[14–16].

Here we show that in 567 community-dwelling older men, higher levels of the biologically active form ($1,25(OH)_2D$) and vitamin D activation and catabolism ratios, but not 25(OH)D, are associated with greater α-diversity. In addition, those men with the highest compared to lowest $1,25(OH)_2D$ and activation ratios are more likely to possess butyrate-producing bacteria that are associated with favorable gut microbial health. These results support the underlying hypothesis that the human gut microbiome and vitamin D metabolism are integrally related.

## Results

**Participant characteristics and vitamin D metabolites.** Men had a mean age of 84 years (SD = 4.1), an average BMI of 27 kg/m², and were overall very physically active (Table 1). Almost 7% of the men reported having taken antibiotics in the past 30 days of the clinic visit. Few men were vitamin D deficient (7.2%; 25(OH)D < 20 ng/ml) and most reported taking some form of vitamin D supplementation (74.8%). Notably, $1,25(OH)_2D$ and $24,25(OH)_2D$ vitamin D levels were significantly correlated with 25(OH)D levels (Pearson correlation coefficient of 0.43, $p < 0.001$ and coefficient of 0.80, $p < 0.001$, respectively). Serum $24,25(OH)_2D$ levels also decreased relative to $1,25(OH)_2D$ levels in vitamin D deficient patients ($p < 0.001$). As might be expected, participants from the clinical site (San Diego) with most annual sun exposure (Fig. 1a) had the highest 25(OH)D levels compared to those with less sun exposure (Minneapolis, Pittsburgh, Portland, and Birmingham) (Fig. 1b)[17]. However, there were no differences in $1,25(OH)_2D$ levels across sites (Fig. 1c), indicating that while sun exposure may affect the storage form of vitamin D, there appears to be less influence on the active hormone.

**Table 1 Baseline characteristics of study participants ($n = 567$)[a].**

| Variables | Mean (SD) or frequency (%) |
|---|---|
| Age | 84.2 (4.1) |
| *Race* | |
| White | 502 (89%) |
| African American | 19 (4%) |
| Asian | 25 (4.4%) |
| Hispanic | 12 (2%) |
| Other | 9 (1.6%) |
| *Site* | |
| Birmingham | 75 (13.2%) |
| Minneapolis | 86 (15.2%) |
| Palo Alto | 68 (12%) |
| Pittsburgh | 92 (16.2%) |
| Portland | 120 (21.2%) |
| San Diego | 126 (22.2%) |
| Body mass index (kg/m²) | 27.0 (3.8) |
| PASE score | 123 (66.9) |
| Self-reported health good/ excellent | 504 (88.9%) |
| *Smoking history* | |
| Never | 216 (38%) |
| Past | 276 (48.7%) |
| Current | 8 (1.4%) |
| *Alcohol use* | |
| Non-drinker | 223 (39%) |
| Up to 13 drinks per week | 315 (57%) |
| ≥14 drinks per week | 27 (4.8%) |
| Vitamin D deficiency (<20 ng/mL) | 40 (7.2%) |
| Vitamin D supplement use | 424 (74.8%) |
| Antibiotic use in past 30 days[b] | 38 (6.7%) |
| Antidepressant use[c] | 51 (9%) |
| Statin use | 315 (56%) |
| Laxative use | 91 (16.1%) |
| Probiotic use | 25 (4.4%) |
| Proton pump inhibitor use | 116 (19%) |
| High resistant starch diet[d] (≥5 g/day) | 72 (13%) |

[a]Range of subjects was 556–567, with the exception of smoking where only 500 men answered this question. Otherwise, if <567, information on a particular item was missing.
[b]Antibiotic use included: Beta-lactam drugs, macrolides, quinolones, aminoglycosides, tetracyclines, antifungals, and others (clindamycin, bacitracin, polymyxin B, mucopirocin). No sulfonamide, no metronidazole, no vancomycin use reported.
[c]Antidepressant use included: Selective serotonin reuptake inhibitors, serotonin norepinephrine-uptake inhibitors, heterocyclic compounds, monoamine oxidase inhibitors, and others (trazodone, bupropion, venlaflaxine, and S-adenosylmethionine). No mirtazapine reported.
[d]Food sources categorized as containing >1 g resistant starch/100 g of food.

**Vitamin D metabolites and α-diversity.** In redundancy analyses, the serum $1,25(OH)_2D$ was the factor that explained the highest proportion of the variance in α-diversity (e.g., bacterial species diversity within an individual) at just over 5% (Fig. 2a). Other non-redundant variables included site, race, recent antibiotic use, antidepressant use, and 25(OH)D. In multiple linear regression analyses adjusted for age, BMI, race, site, antibiotic use, antidepressant use, physical activity score, season of visit, and total starch intake, there was greater α-diversity with higher $1,25(OH)_2D$ levels ($p = 7.23 \times 10^{-7}$) (Fig. 3). Similarly, α-diversity was higher in men with higher levels of $24,25(OH)_2D$ ($p = 0.02$) and those with higher ratios of activation ($p = 0.0002$) and catabolism ($p = 0.003$). In this study cohort, 6.7% reported recent antibiotic use in the past 30 days and α-diversity was significantly reduced in these men (Fig. 4a). However, even after adjustment for antibiotic use, the significant association between active $1,25(OH)_2D$ and gut microbial α-diversity persisted.

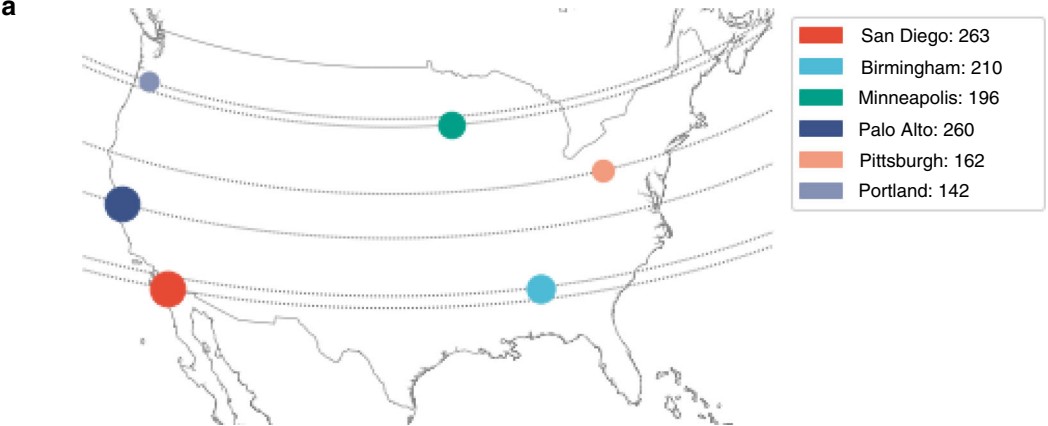

*Data through 2018, demonstrate the number of clear or partially cloudy days
(https://www.ncdc.noaa.gov/ghcn/comparative-climatic-data)

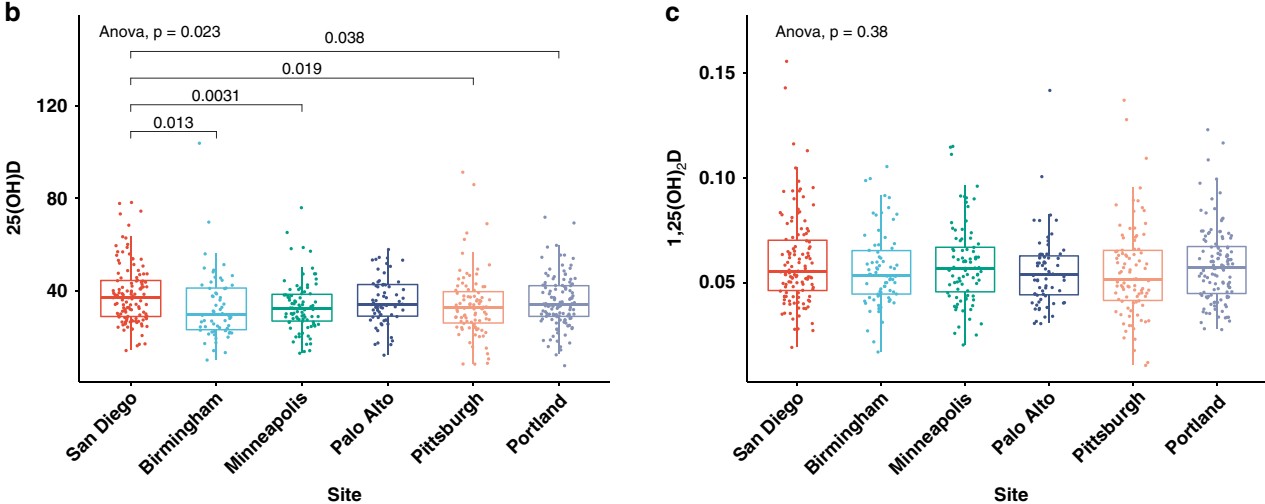

**Fig. 1 25(OH)D levels vary with site and sun exposure, but 1,25(OH)₂D levels do not follow this association. a** San Diego has more sunny days and a corresponding higher vitamin 25(OH)D level than Birmingham, which lies on similar latitude but enjoys fewer sunny days based on National Oceanic and Atmospheric Administration (NOAA) comparative climatic data. **b** Significant differences in 25(OH)D levels according to study site (box plots indicate median (middle line), 25th, 75th percentile (box) and each single dot represents a sample, with sample size $n = 125$ for San Diego, 64 for Birmingham, 86 for Minneapolis, 68 for Palo Alto, 92 for Pittsburgh, and 120 for Portland; $p$-values from two-sided $t$-test for pairwise comparisons and ANOVA test for all-site comparisons). **c** No significant differences in 1,25(OH)₂D levels according to study sites (box plots indicate median (middle line), 25th, 75th percentile (box) and each single dot represents a sample, with sample size $n = 125$ for San Diego, 75 for Birmingham, 86 for Minneapolis, 68 for Palo Alto, 92 for Pittsburgh, and 120 for Portland; $p$-values from two-sided $t$-test for pairwise comparisons and ANOVA test for all-site comparisons). Source data are provided as Source data file: "sourcedata_Fig. 1.txt".

**Vitamin D metabolites and β-diversity**. In redundancy analyses using unweighted UniFrac, 1,25(OH)₂D explained the highest proportion of variation in microbial β-diversity (~2%), a measure that indicates bacterial species differences between individuals. In comparison with α-diversity analysis, additional factors, including total starch intake, statin use, age, physical activity, and PPI (proton pump inhibitor) use were identified as non-redundant covariates for β-diversity (Fig. 2b). In β-diversity testing results with PERMANOVA after BH-FDR correction and consistent with redundancy analysis results, most non-redundant covariates retained statistical significance with the exceptions of 25(OH)D ($q = 0.32$) and age ($q = 0.058$). Based on the clinical definition of vitamin D deficiency (25(OH)D < 20 ng/ml), we also examined vitamin D 25(OH)D as a dichotomous variable and found that it made no difference in the results ($q = 0.503$). Thus, neither treating 25(OH)D as a continuous nor categorical variable had a significant impact on β-diversity in our study sample. In contrast,

differences in serum levels of 1,25(OH)₂D ($q = 0.004$), 24,25(OH)₂D ($q = 0.011$), the vitamin D activation ratio ($q = 0.004$), and the vitamin D catabolism ratio ($q = 0.004$) separated subjects into statistically significant clusters within the β-diversity distribution (Fig. 5). Covariates including clinical site ($q = 0.004$), race ($q = 0.004$), total starch intake ($q = 0.004$), physical activity score ($q = 0.013$), and alcohol intake ($q = 0.035$) also defined significant clusters based on β-diversity. Of medications analyzed, subjects who reported antibiotic use ($q = 0.006$) (Fig. 4), antidepressant use ($q = 0.006$), statin use ($q = 0.013$) or PPI use ($q = 0.026$) exhibited significant differences in β-diversity distribution, while vitamin D supplementation ($q = 0.545$), probiotic use ($q = 0.464$), laxative use ($q = 0.205$), and antihistamines ($q = 0.545$) did not significantly distinguish subjects into different clusters based on β-diversity. Similarly, tobacco use ($q = 0.236$) was not significant. Overall, apart from known correlates such as race and geographic locations, measures of vitamin D metabolic flux were

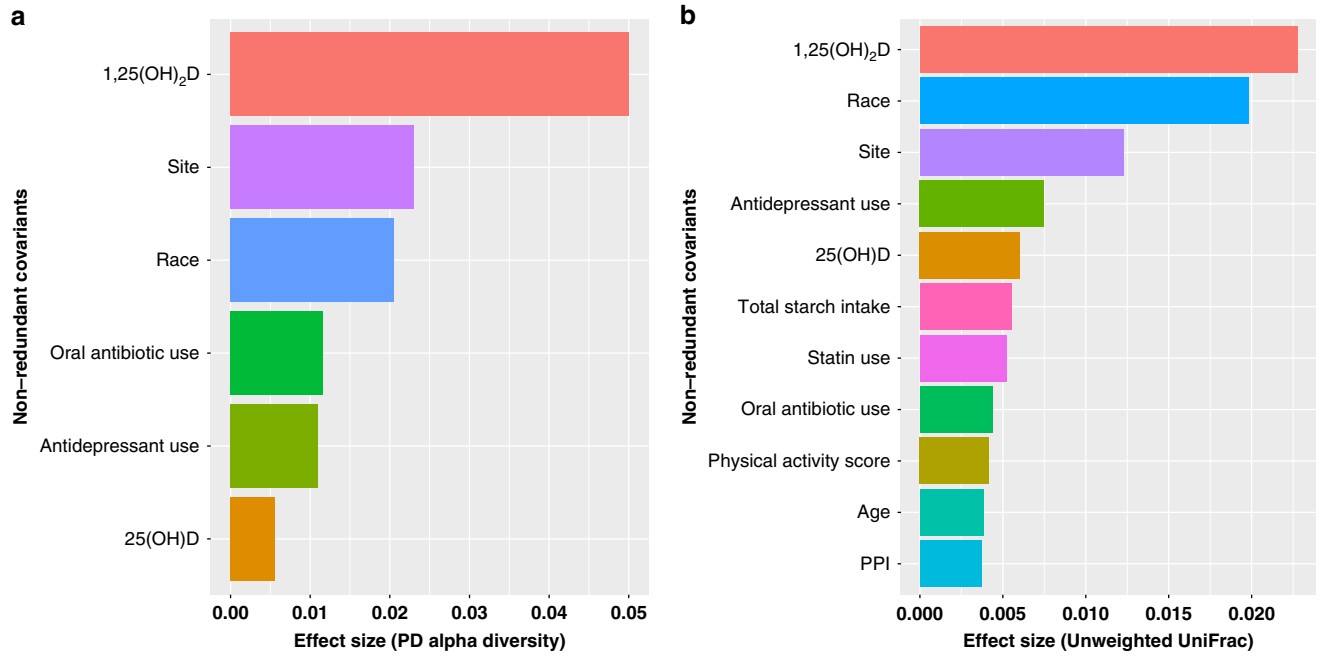

**Fig. 2 1,25(OH)₂D levels explain the highest proportion of the variance in α-diversity and β-diversity.** Forward stepwise redundancy analysis (RDA) of non-redundant variables explaining variation in α-diversity (**a**) and β-diversity (**b**) among candidate covariates that included age, site, race, body mass index, vitamin D metabolites, vitamin D activation/catabolism ratios, health behaviors, medications, and dietary intake of resistant starches. Source data are provided as Source data file: "sourcedata_figure2a.txt and sourcedata_figure2b.txt".

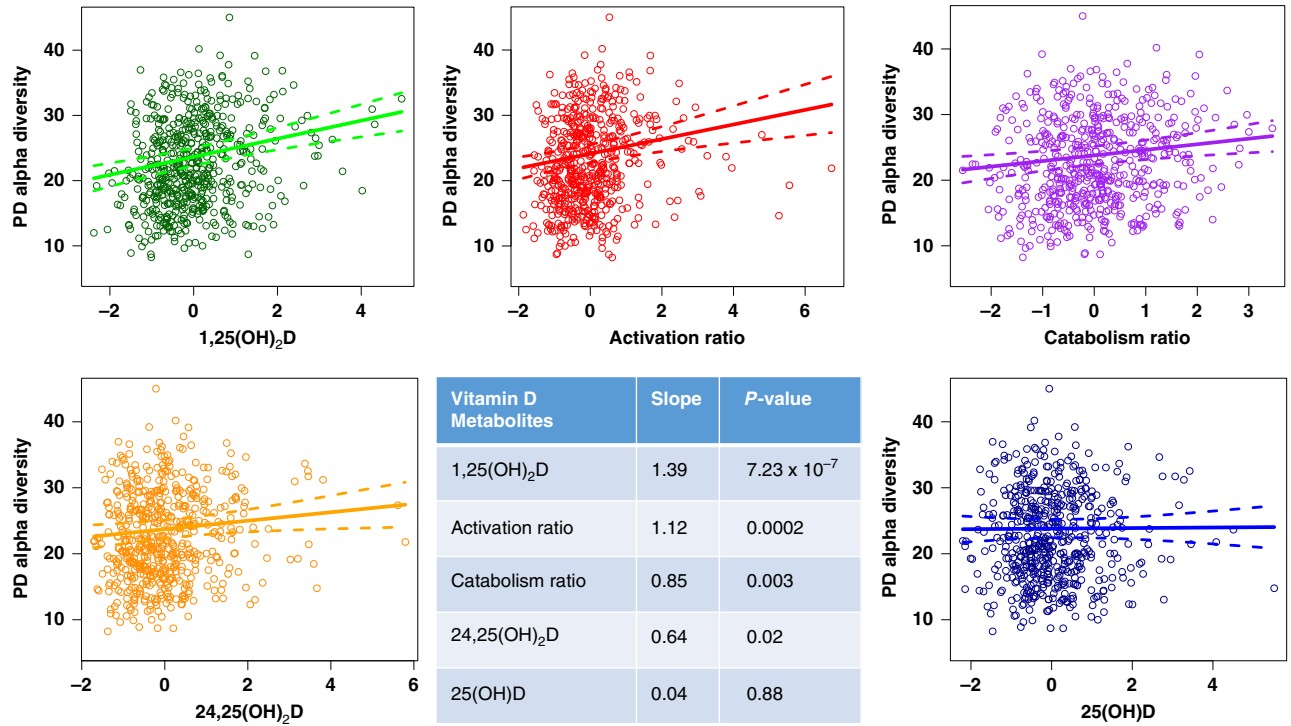

**Fig. 3 Greater α-diversity is associated with higher 1,25(OH)₂D levels and larger vitamin D activation and catabolism ratios.** Multiple linear regression of each of the vitamin D metabolites and their activation/catabolism ratios in association with microbial α-diversity. Covariates are set at their mean for continuous variables and level with largest sample size for categorical variables. Slopes are the estimated slope coefficients corresponding to each vitamin D metabolites in their multiple linear regression models, and p-values are obtained from the ANOVA F-test in multiple linear regression. Source data are provided as Source data file: "sourcedata_figure3.xlsx".

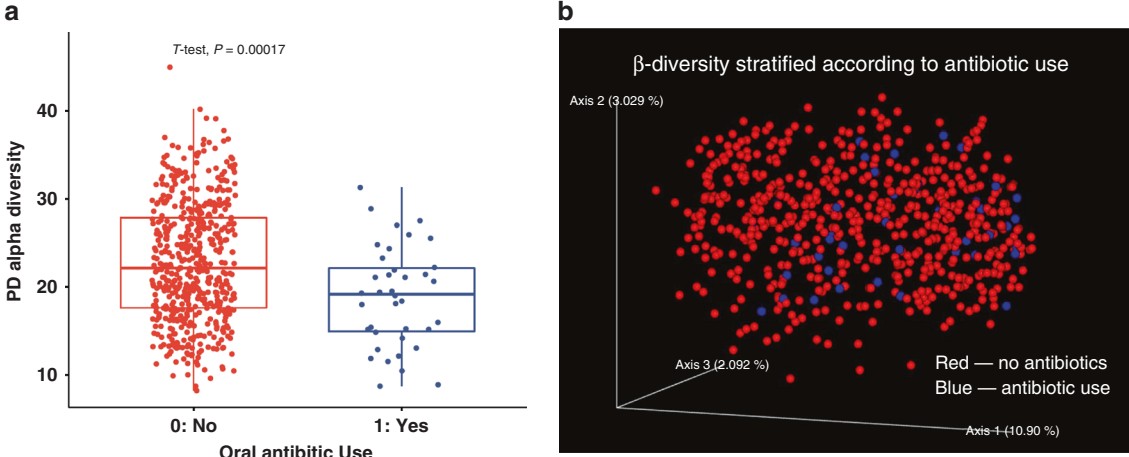

**Fig. 4 Antibiotic use correlates with reduced α-diversity and defines significant β-diversity clusters. a** Reduced α-diversity in patients who have taken antibiotics within in the last 30 days (box plots indicate median (middle line), 25th, 75th percentile (box) and each single dot represents a sample, with sample size $n = 528$ for subjects with no oral antibiotic use and $n = 38$ for subjects with oral antibiotic use; p-value obtained from two-sided t-test). **b** Unweighted UniFrac PCoA plot showing significant difference in β-diversity associated with antibiotic use (q-value = 0.006 from PERMANOVA test after BH-FDR correction). Source data are provided as Source data file: "sourcedata_fig4a.txt and sourcedata_fig4b_fig5.txt".

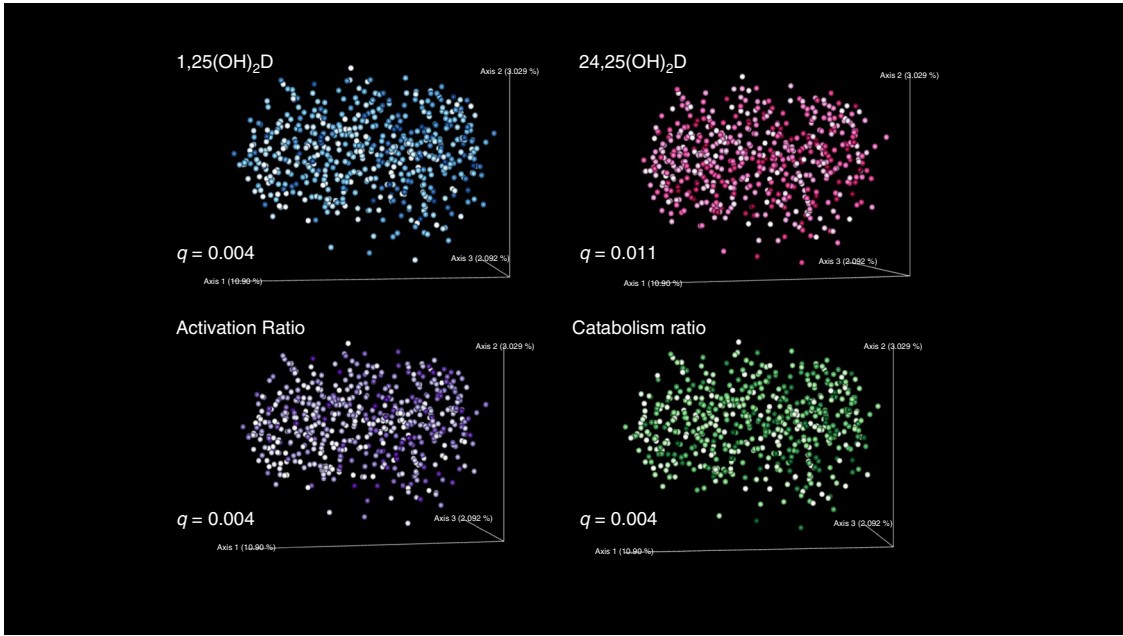

**Fig. 5 Vitamin D metabolites correlate with clusters in β-diversity distribution.** Vitamin D activation ratio ($q = 0.004$), and the vitamin D catabolism ratio ($q = 0.004$) separated subjects into statistically significant clusters within the β-diversity distribution Unweighted UniFrac β-diversity PCoA plots of vitamin D metabolites. β-diversity is significantly stratified according to 1,25(OH)$_2$D, 24,25(OH)$_2$D, vitamin D activation ratio, and vitamin D catabolism ratio based on PERMANOVA test after BH-FDR correction. Darker colors correspond to higher metabolite values. Source data are provided as Source data file: "sourcedata_fig4b_fig5.txt".

remarkably associated with microbial β-diversity in unweighted UniFrac and PERMANOVA testing.

**Vitamin D metabolites and specific taxonomies.** In random forest analyses with 5-fold cross-validation of bacterial gene sequences, 12 unique stool sub-Operational Taxonomic Units (sOTUs) were identified to correlate with vitamin D metabolism (Table 2). Six sOTUs were associated with 1,25(OH)$_2$D, of which all were from the Firmicutes phlyum, Clostridia class, and Clostridia order and recognized to be producers of butyrate. Of the eight sOTUs identified with the ratio of activation, seven were

from the Firmicutes and one from the Lentisphaerae phylum, with all of the Firmicutes being further classified into the Clostridia class and Clostridiales order. Beyond that, two were from Ruminococcaceae family, one of which was identified to the *Oscillospira* genus. The remaining Firmicutes sOTUs were from the Lachnospiraceae and Victivallaceae family and *Mogibacterium* genus. The only sOTU from the Lentisphaerae phylum belonged to the family Victivallaceae. For all but three sOTUs, the Firmicutes phylum was positively associated with higher levels of 1,25(OH)$_2$D and the ratio of activation, while the direction of the association was negative in the *Oscillospira*, *Blautia*, and *Anaerotruncus* genus (Table 2).

**Table 2 sOTUs identified via random forest as important for classifying 1,25(OH)$_2$D, and/or the ratio of activation, ranked by importance score.**

| Taxa | 1,25(OH)$_2$D[a] | Ratio of activation[a] | Importance score[b] |
|---|---|---|---|
| P-Firmicutes, C-Clostridia, O-Clostridiales, F-Ruminococcaceae, G-Oscillospira | Negative | Negative | 0.0023, 0.0078 |
| P-Firmicutes, C-Clostridia, O-Clostridiales, F-Ruminococcaceae, G-Ruminococcus | Positive | N/A | 0.004 |
| P-Firmicutes, C-Clostridia, O-Clostridiales, F-Lachnospiraceae, G-Blautia | Negative | Negative | 0.0022, 0.0086 |
| P-Firmicutes, C-Clostridia, O-Clostridiales, F-Lachnospiraceae, G-Coprococcus, S-Catus | Positive | N/A | 0.0026 |
| P-Firmicutes, C-Clostridia, O-Clostridiales, F-Lachnospiraceae, G-Blautia, S-Obeum | Positive | N/A | 0.0028 |
| P-Firmicutes, C-Clostridia, O-Clostridiales, F-Ruminococcaceae, G-Anaerotruncus | Negative | N/A | 0.0022 |
| P-Firmicutes, C-Clostridia, O-Clostridiales, F-Ruminococcaceae | N/A | Positive | 0.0155 |
| P-Firmicutes, C-Clostridia, O-Clostridiales, F-Lachnospiraceae | N/A | Positive | 0.0146 |
| P-Firmicutes, C-Clostridia, O-Clostridiales, F-Mogibacteriaceae, G-Mogibacterium | N/A | Positive | 0.0131 |
| P-Lentisphaerae, C-Lentisphaeria, O-Victivallales, F-Victivallaceae | N/A | Positive | 0.0078 |
| P-Firmicutes, C-Clostridia, O-Clostridiale | N/A | Positive | 0.0149 |
| P-Firmicutes, C-Clostridia, O-Clostridiales, F-Lachnospiraceae, G-Coprococcus | N/A | Positive | 0.0233 |

[a]Direction of associations between taxa identified by random forest and each vitamin D metabolite determined by Spearman rank correlation coefficients.
[b]Significance of correlation decided by Spearman's p-value with BH-FDR correction.

## Discussion

We report robust correlations between the vitamin D metabolites, 1,25(OH)$_2$D and 24,25(OH)$_2$D, and the gut microbiome in 567 older men representing six geographic sites across the United States. Those men with higher levels of 1,25(OH)$_2$D had greater α-diversity, even after adjusting for previously characterized determinants of microbial diversity including age, geographical origin, race, PPI, and antibiotic use[18]. Notably, 1,25(OH)$_2$D levels exhibited a much greater effect size on α-diversity than these other covariates. Similar findings were found with 1,25(OH)$_2$D and β-diversity.

When examining those with the highest levels of each of the vitamin D measures compared to those with the corresponding lowest levels, men with the highest 1,25(OH)$_2$D and/or activation ratios were more likely to harbor specific genera known to be butyrate producers or to provide the substrate for bacteria that produce butyrate.

Strikingly, 25(OH)D was not strongly associated with any microbiota measures, whether assessing α-diversity, β-diversity, or specific sOTUs. Serum 25(OH)D is the preferred clinical measure used because it is representative of overall body stores of vitamin D; however, our results suggest that it is the regulation of VDM, reflected by the active hormone and metabolic ratios rather than body stores that may have the most health implications. The most recent evidence to date does not support global vitamin D supplementation in community-dwelling adults in the general population unless it is targeted at those with pre-existing skeletal disease and 25(OH)D levels in the deficient range[19]. Further complicating the clinical understanding of whether there is an indication for prescribing vitamin D supplementation, the largest randomized controlled trial of vitamin D3 2000 IU versus placebo daily to prevent cardiovascular disease or cancer in over 25,000 older adults reported no benefit[11].

Notably, in our study, most men (74%) reported taking some vitamin D supplementation, and only 7% met the definition of being vitamin D deficient. As expected, higher levels of the prohormone 25(OH)D were strongly associated with greater 24,25(OH)$_2$D production. Endocrine feedback regulates active hormone 1,25(OH)$_2$D levels while shifting surplus 25(OH)D and 1,25(OH)$_2$D toward the catabolic 24-hydroxylase pathway. Accordingly, a higher proportion of vitamin D was activated to 1,25(OH)$_2$D in vitamin D deficient patients while vitamin D catabolism ratios were greater in patients with adequate vitamin D stores (Fig. 6). Appropriate with the predicted tenets of feedback control of endocrine vitamin D metabolism, lower prohormone availability thus appropriately favors active hormone production over catabolism. The positive association between diversity metrics and vitamin D activation and catabolism ratios suggests that physiologically normal vitamin D flux is more likely to occur in individuals with healthy microbiomes. In this model, the prohormone can be activated more quickly to increase 1,25(OH)$_2$D availability and deactivated more quickly to keep homeostatic mechanisms responsive and avoid vitamin D toxicity in the host.

The significant correlation between amplified vitamin D activation and greater individual stool microbial diversity supports the idea that increased microbial diversity reflects a healthy state. In the healthy colon, butyrate-producing bacteria including Firmicutes provide enterocytes with short-chain fatty acids for energy production[20]. Indeed, loss of butyrogenic bacteria and decreased overall diversity have both been associated with nosocomial diarrheal illness[21]. Of the specific sOTUs identified in our study, 92% belong to the Firmicutes phylum, most of which were positively correlated with increased levels of 1,25(OH)$_2$D and the vitamin D activation ratio in the expected directions. Thus, in totality, our study findings support previous study assertions of a dynamic interplay between the active vitamin D metabolites and butyrate-producing bacteria[5,22].

Our study cannot determine whether it is enhanced vitamin D signaling that predisposes to the predominance of butyrate-producing Firmicutes in the colon or vice versa. We hypothesize that butyrate-producing colonic microbiota stimulate the local production of 1,25(OH)$_2$D by colon-resident immune cells (e.g., dendritic cells)[23,24]. Further stipulating that the gut microbiota may affect host metabolism, researchers studied germ-free (GF) mice to determine the effects of microbes on host vitamin D physiology[25]. In 20 GF male and female C57Bl/6 wild-type mice, prior to conventionalization (CN), serum 25(OH)D, 1,25(OH)$_2$D, and 24,25(OH)$_2$D levels were low and the mice exhibited hypocalcemia. Within 2 weeks of CN with microbiota, some of which included butyrate-producing bacteria, there were significant increases in all vitamin D metabolites with re-instated calcium homeostasis.

Additional studies, including a recent review, suggest that intact vitamin D signaling is important for a healthy gut microbiome, with bi-directional signaling between bacteria and colonic epithelium[26]. For example, it has also been shown that butyrate, in combination with 1,25(OH)$_2$D in the conditioning media,

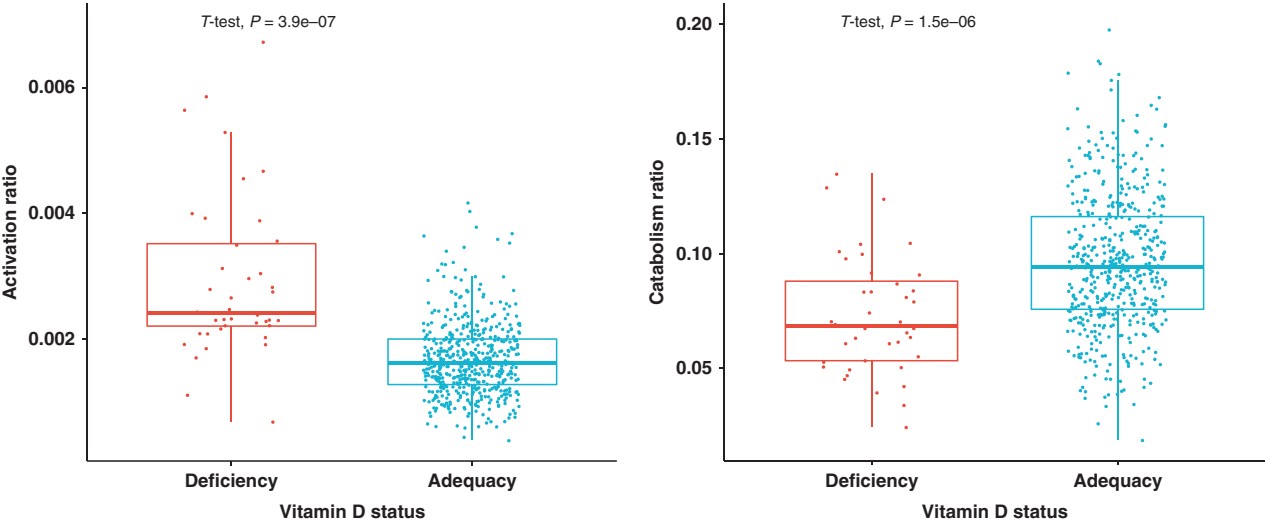

**Fig. 6 Vitamin D metabolic ratios vary with differences in 25(OH)D status.** Vitamin D deficiency correlates with significantly higher vitamin D activation ratio (**a**), while the catabolism ratio is increased in vitamin D adequacy (**b**) (box plots indicate median (middle line), 25th, 75th percentile (box) and each single dot represents a sample, with sample size $n = 40$ for subjects with vitamin D deficiency and $n = 515$ for subjects with vitamin D adequacy; $p$-values obtained from two-sided $t$-test). Source data are provided as Source data file: "sourcedata_fig6.txt".

leads to upregulation of VDR expression in cultured colonic epithelial cells[22]. These study findings delineate a role in the microbiome as directly influencing vitamin D metabolism.

Studies have demonstrated that mice with disrupted vitamin D metabolism had intestinal dysbiosis[3], while studies in humans suggest that manipulation of either the gut microbiome or VDM may have a favorable health impact[6,27]. For example, in studies of vitamin D receptor (VDR) and CYP27B1-hydroxylase deficiency (which impairs 1,25(OH)$_2$D formation) knockout mice, loss of vitamin D signaling results in increased levels of Bacteroides and Proteobacteria, with fewer normal Firmicutes including Ruminococcaceae, Lachnospiraceae, and Lactobacillaceae[5]. Several human studies have manipulated the gut microbiome by introducing lactobacillus and have reported increased VDR expression in human cell cultures[4,6,28–30]. In healthy humans, a small study of high dose vitamin D supplementation suggests that vitamin D in doses ranging from 4400 to 9800 IU daily (for a 70 kg person) given over 8 weeks did not change the composition of the stool microbiome[31]. However, in patients with cystic fibrosis (chronic disease associated with gastric malabsorption) who were vitamin D insufficient (25(OH)D < 30 ng/mL) and supplemented with either vitamin D3 50,000 IU or placebo weekly for 12 weeks, those who received vitamin D had experienced an enrichment of Lactococcus that is associated with better gut health[27]. These human studies suggest that vitamin D supplementation may only provide benefit in those who are physiologically deficient and are not helpful in those who are already sufficient. Overall, the interplay between vitamin D signaling and the microbiome remains elusive, but literature suggests potential for a pathogenic cascade in which vitamin D deficiency and dysbiosis produce synergistic insults that exacerbate microbial dysregulation and systemic disease.

Our study has some limitations. First, it is cross-sectional, so we are unable to determine causality or direction of the found associations. Second, we are limited by taxonomic classifications based on 16S rRNA amplicon profiling rather than comprehensive shotgun metagenomic sequencing data. Third, we studied only older predominantly white men, so our results may not apply to other populations. Our study also has notable strengths. It is a large study of 567 older men, recruited from six U.S. geographically diverse areas who have been extensively phenotyped in terms of

body size, health behaviors, and reported medication use. We used state-of-the-art pipeline processing of stool samples and demonstrated robust correlations between serum vitamin D metabolite levels and the gut microbiome that were reproduced across analyses of α and β-diversity. Finally, we employed a supervised machine learning approach, random forest classification, to predict the vitamin D metabolite status from the microbial composition. Based upon the microbiome sequence datasets, this method allows for classification of unlabeled samples with some degree of accuracy (all AUC ROC values reported were over 0.70), highlighting the strength of the relationship between the gut microbiome and vitamin D metabolism.

In summary, we provide strong evidence of important interactions between host vitamin D signaling and the health of the gut microbiome in older men. Co-localized expression of the CYP27B1- and CYP24A1-24-hydroxylases in the gut may be potentiated or inhibited by the types of microbiota present. The consistency and robustness of results in finding active vitamin D metabolites associated with more favorable gut microbial diversity, including specific microbiota that are known butyrate producers, provide potential targets for intervention, whether through dietary modification and/or vitamin D supplementation in clinically appropriate populations.

## Methods

**Study participants.** Originating in 2000–2002, the Osteoporotic Fractures in Men Study (MrOS) included 5994 men aged 65 years and older from six U.S. clinical sites, with continued follow-up involving four full in-clinic examinations through 2016. The study design and recruitment have been previously described[32]. For the current study, 1841 surviving participants attended visit 4 (2014–2016). Starting in March 2015, 1328 men were invited to provide stool specimens, of which 982 agreed. The Institutional review boards of the six participating institutions approved the study protocol, and written informed consent was obtained from all participants.

**Stool collection and processing.** In March–April 2015, MrOS study began the collection of stool specimens for microbiome analysis at six clinical U.S. sites. Participants at all 6 MrOS sites collected their fecal samples using the OMNIgene-GUT stool/feces collection kit (OMR-200, DNA Genotek, Ottawa, Canada) and then mailed their samples directly to the Portland site for initial processing. The first 599 samples collected were then shipped on dry ice overnight to Baylor College of Medicine in Houston, Texas, for 16S ribosomal RNA sequencing[33], which follows the EMP protocol[34]. We targeted and amplified the V4 region of the 16S rRNA gene by PCR using the 515F and 806R primers (a complete list of all primers

used are shown in Supplementary Table 1)[35,36]. The V4 paired-end sequencing[35] was performed using an Illumina MiSeq with $2 \times 250$ cycles according to manufacturer's protocols. Raw V4 sequence reads were demultiplexed using Illumina's bcl2fastq software version 2.20.0.433. Primers were trimmed via cutadapt 1.18[37] uploaded to Qiita[38] and quality controlled using the defaults. Forward reads were trimmed to the first 150 nucleotides. The primary feature table was generated using Deblur 1.1.0[39]. Next-generation sequencing data are subject to sequencing errors at a rate of 0.1% per nucleotide[40]. These sequencing errors falsely increase diversity estimates and lead to inaccurate/spurious taxon identification in large genomic studies. To counter this problem, we utilized the Deblur algorithm developed by the Knight lab with a phylogenetic tree built using SEPP 4.3.5[39,41]. SEPP takes advantage of a highly curated ribosomal full length reference phylogeny—here Greengenes 13.8—and extends it by placing obtained short-read fragments into the reference to form an insertion tree[42]. Together, Deblur and SEPP minimize sequence errors and phylogenetic bias while producing quantitative descriptions of the microbiota present in each stool sample. The taxonomic label of each sOTU was assigned by traversing the insertion tree from a high-resolution sOTU toward the root until encountering a known OTU.

**Vitamin D measurements**. Fasting serum concentrations of three vitamin D metabolites: 25(OH)D, 1,25(OH)$_2$D, and 24,25(OH)$_2$D, were analyzed from 567 men who had serum stored at −80 °C until assayed. Specifically, the measure of 1,25(OH)$_2$D was 1α,25(OH)$_2$D$_3$ and the measure of 24,25(OH)$_2$D was of 24,25 (OH)$_2$D$_3$. Measures were performed using liquid chromatography coupled to tandem mass spectrometry (LC-MSMS) at University Hospital Leuven/Katholieke Universiteit, Leuven, Belgium.

The combined measurement of 1α,25(OH)$_2$D$_3$ and 24,25(OH)$_2$D$_3$ was performed by the LC-MSMS method published by Cassetta et al.[43], later modified as described by Vanderschueren et al. and Cools et al.[44,45], and adapted to further increase specificity for isobaric vitamin D metabolites[46]. The method consists of protein precipitation of 200 μL serum with 400 μL methanol containing deuterated internal standards (d6-1α,25(OH)$_2$D$_3$ and d6-24,25(OH)$_2$D$_3$), followed by injection of 180 μL supernatant (SIL-20AC autosampler (Shimadzu) in the 2-dimensional chromatographic LC system (Poros R1/10 (4.6 × 50 mm) (Applied Biosystems) as first dimension kept at 30 °C (in CTO-20AC column oven (Shimadzu)) and three Onyx Monolithic C18 (3.0 × 100 mm) (Phenomenex) in series at room temperature as second dimension)). The eluents were pumped by four LC-20AD pumps (Shimadzu) and a Rheodyne 10 port/2 position valve was used to switch flows. After injection, two pumps generated a 3 min gradient from 98% water and 2% of a methanol/acetonitrile (75/25 v/v) mixture to 100% methanol/acetonitrile over the first dimension at a flow rate of 1.5 mL/min, from 3 to 4.5 min the Rheodyne valve was switched and the flow of 100% methanol/acetonitrile over the first dimension (reduced to 0.25 mL/min) was combined with the 100% 0.5 M lithium acetate in water flow (0.25 mL/min) (third pump) to generate a combined flow of 0.5 mL/min over the Onyx Monolithic columns. After re-switching of the Rheodyne valve, a 9 min gradient starting from 30% 0.5 mM lithium acetate in water and 70% 0.5 mM lithium acetate in methanol to 100% 0.5 mM lithium acetate in methanol was applied over the three Onyx Monolithic columns at 0.5 mL/min (by the third and fourth pump), meanwhile the first dimension was re-equilibrated. Afterward, 100% 0.5 mM lithium acetate was kept for 2.5 minutes over the second dimension, resulting in a total run time of 16 min. The mass spectrometer (AB Sciex 5500 QTRAP) was operated in electrospray positive mode (ion spray voltage 5500 V, source temperature 550 °C, declustering potential 180 V, entrance potential 10 V, and curtain gas 35 psi). Lithium adducts of the dihydroxylated vitamin D metabolites and isotopes were monitored. On the mass spectrometer, transitions between 423 > 369 and 429 > 393 were monitored for 1α,25(OH)$_2$D$_3$ and d6-1α,25(OH)$_2$D$_3$, respectively. Transitions between 423 > 467 and 429 > 473 were monitored for 24,25(OH)$_2$D$_3$ and d6-24,25(OH)$_2$D$_3$, respectively. Retention time was 13.3 min for 1α,25(OH)$_2$D$_3$ and 12.6 min for 24,25(OH)$_2$D$_3$.

Calibration curves ranging from 6.5 to 250 pg/mL for 1α,25(OH)$_2$D$_3$ and 0.2 to 55 ng/mL for 24,25(OH)$_2$D$_3$ were prepared by spiking ethanolic standard solutions, calibrated by UV absorbance at 264 nm using a molar absorbance of 18,300, in surrogate matrix (bovine serum albumin dissolved in physiological water at 60 g/L containing 0.2% serum with very low concentration 1α,25(OH)$_2$D$_3$). Calibrators were run in every batch and a repeat serum sample at relevant physiological concentration was included in every batch. The method showed a between-run imprecision of 6.7% at 40 pg/mL for 1α,25(OH)$_2$D$_3$ and 7.6% at 2.0 ng/mL for 24,25(OH)$_2$D$_3$, the lower quantification limit was 10 pg/mL for 1α,25(OH)$_2$D$_3$ and 0.2 ng/mL for 24,25(OH)$_2$D$_3$ (signal-to-noise at least 10), and the method was linear for the calibration interval (up til 250 pg/mL for 1α,25(OH)$_2$D$_3$ and 55 ng/mL for 24,25(OH)$_2$D$_3$). Interference from potentially relevant dihydroxylated isobaric vitamin D metabolites was extensively examined[46]. The median concentration of 1α,25(OH)$_2$D$_3$ and 24,25(OH)$_2$D$_3$ in the samples of this study was 56 pg/mL and 3.2 ng/mL, respectively. Study samples were included in the routine patient runs for 1α,25(OH)$_2$D$_3$ analysis ($n = 57$); quality control results for the runs were within 2 standard deviations of the preset mean, except for 1 which was between 2 and 3 standard deviations. The laboratory participates in the Vitamin D External Quality Assessment Scheme (DEQAS; 4 rounds of 5 samples every year) and showed acceptable performance based on the criteria set by the organization.

The method for measurement of serum 25(OH)D has not been published earlier in a peer-reviewed journal. Sample pretreatment consisted of adding 200 μL protein precipitation reagent (a mixture of 0.3 M zinc sulfate in water with methanol (20:80 (v/v)) containing 8 ng/mL d6-25(OH)D$_3$) to 50 μL serum in a 96-well plate. The plate was shaken on an orbital 96-well plate shaker (Provocell® shaking micro incubator, Esco) for 420 s at 7.5 × g, the incubation period was 3000 s and the final shake was 60 s at 7.5 × g. Thereafter, the plate was centrifuged using a Rotina 380R cooled centrifuge (Hettich, Tuttlingen, Germany) with swing-out rotors for 96-well plates at 3954 × g for 10 min at 4 °C. After centrifugation, the plate was capped with a pre-slit 96-well cap (silicone cap, Agilent, Santa Clara, USA) and directly transferred to the liquid chromotagraphy autosampler. MassChrom 3PLUS1 multilevel calibrator set from Chromsystems (Gräfelfing, Germany) (with National Institute for Standards and Technology [NIST] 972a traceable serum values) was used for calibration. Calibration was performed in every run.

The method was run on a AB Sciex 5500 QTRAP mass spectrometer connected with a Shimadzu chromatographic system consisting of two Nexera XR LC-20ADxr units (high pressure mixing gradient pumps) and a LC-20AD unit (loading pump), two DGU-20A5R (degasser units), a Nexera XR SIL-20AC injector unit, a CBM-20A communications module, a CTO-20AC column oven containing one 10 port valve (FCV-12AH), one column selector valve (FCV-14AH) and one manifold, and a compartment (ambient temperature) containing two FCV-32AH valve units (6 port switching valves). The method consisted of an online cleanup step using a Strata C8 (20 × 2.0 mm, 20 μm, Phenomenex, Torrance, USA) online extraction cartridge at ambient temperature. After loading and cleaning, the analytes were eluted by back-flushing. A Kinetex F5 (100 × 3.0 mm, 2.6 μm, Phenomenex) equilibrated at 45 °C was used as the analytical column. The injection volume was 20 μL and the autosampler was kept at 6 °C. The solvents, gradients, flows, and switching scheme is presented in Supplementary Table 2. Atmospheric-pressure chemical ionization (APCI) positive mode was used at 400 °C. Mass transition settings are summarized in Supplementary Table 3.

The retention time of 25(OH)D$_3$ and d6-25(OH)D$_3$ was 3.0 min. The retention time of 25(OH)D$_2$ was 3.2 min. The chromatographic method separated out the 3-hydroxy-epimers of 25(OH)D$_3$ and 25(OH)D$_2$, which are isobaric metabolites that can be present in young children at physiologically relevant concentrations as compared to 25(OH)D; 25(OH)D was the sum of 25(OH)D$_3$ and 25(OH)D$_2$. In adult serum, as encountered here, the 3-hydroxy epimers of 25(OH)D$_3$ and 25(OH)D$_2$ are typically detected at physiologically irrelevant concentrations.

QC samples were prepared from pooled leftover serum. Two QC samples were included in every batch. Study samples were run in 7 batches with a between-run imprecision of 5.6% at 28.9 ng/mL 25(OH)D. The median concentration of 25(OH)D in the samples of this study was 34.2 ng/mL. The method was linear from 2.0 to 80 ng/mL for 25OHD$_3$ and from 0.7 to 47 ng/mL for 25(OH)D$_2$. Limit of quantification was 2.0 ng/mL for 25(OH)D$_3$ and 0.7 ng/mL for 25(OH)D$_2$ (signal-to-noise at least 10). The laboratory participates in the vitamin D external quality Assessment Scheme (DEQAS; 4 rounds of 5 samples every year) and showed acceptable performance based on the criteria set by the organization.

**Other measurements**. Covariates included age, race, body mass index (kg/m$^2$), smoking status, alcohol intake, self-rated health, self-reported physical activity (assessed via the Physical Activity Scale for the Elderly)[47], season of sample collection, and medication use including (1) antibiotics in the past 30 days; (2) vitamin D3; (3) antidepressants; (4) probiotics; (5) laxatives; (6) statins; (7) antihistamines; and (8) PPI. Estimated dietary intake of resistant starches was obtained from 80 food sources derived from the Food Frequency Questionnaire and defined as containing >1 g of resistant starch/100 g of food.

**Statistical analysis**. Analysis of microbiome communities was conducted in QIIME 2 (Quantitative Insights Into Microbial Ecology) (2020.2 distribution) to calculate microbial diversity and population frequency[48]. Microbial populations were grouped using principal coordinate analysis according to their association with levels of 25(OH)D, 1,25(OH)$_2$D, and 24,25(OH)$_2$D levels as well as activation and catabolism ratios. The ratios of vitamin D metabolites included: 1,25(OH)$_2$D/25(OH)D as a measure of activation and 24,25(OH)$_2$D/25(OH)D as a measure of catabolism[14,15]. Initial analyses included all 5 vitamin D measures as continuous variables, but a dichotomous variable to represent clinically defined vitamin D deficiency (25(OH)D < 20 ng/mL) was also used. Microbial α-diversity was quantified using Faith's phylogenetic diversity (PD), and non-redundant covariates were identified using a forward stepwise redundancy analysis (RDA) with the rda and ordiR2step functions in vegan package in R 3.6.1[49]. RDA is a direct gradient analysis technique which summarizes the linear relationships between components of response variables that are "redundant" with (i.e., explained by) a set of explanatory variables. Automatic forward procedure was applied to RDA to select a subset of explanatory variables (i.e., non-redundant covariates) with a two-step procedure to prevent a highly inflated type I error and an overestimation of the amount of explained variance[50]. First, a global test using all explanatory variables was done prior to forward selection. Second, forward selection was carried out with permutation test (1000 permutations) using two stopping criteria: (1) the usual significance level alpha (prespecified as 0.05) and (2) the adjusted coefficient of multiple determination ($R^2$). This analysis estimates the linear cumulative and

independent effect size (based on adjusted $R^2$) of each non-redundant covariate on microbiome composition variation[18]. Beta diversity analysis included principal coordinates analysis (PCoA) on unweighted UniFrac and PERMANOVA tests with multiple testing correction of the Benjamini—Hochberg (BH) FDR procedure, to identify covariates that explain a significant variation of β-diversity[51,52]. All the BH-FDR corrected p-values are denoted as q-values. Because distance-based redundancy analysis with automatic stepwise model selection procedure that prevents inflated type 1 error is not available when applying RDA to β-diversity, we used the first 10 principal coordinates from PCoA on unweighted UniFrac that provided a low dimensional approximation to the original β-diversity distance matrix. To ensure that the usage of only 10 principal coordinates did not bias our results, we also repeated the same analysis by using all the principal coordinates. The same set of 11 non-redundant covariates was found to be significant as in the analysis with 10 principal coordinates.

PD was chosen because it is the only metric that assesses phylogenetic diversity. Multiple linear regression (MLR) was used to explore the relationship between PD and the five vitamin D metabolite measurements, while accounting for the other explanatory variables and adjusting for possible confounders. The MLR analysis was performed with a two-step approach separately for each vitamin D metabolite measure: first, we used stepwise backward selection with ANOVA type II test to select the confounding variables that significantly affect PD (with a liberal threshold of p-value set at 0.2); then we added the vitamin D measure of interest to the backward selection model to derive the final reported model, where the estimated coefficient for each vitamin D measure is interpreted as the average change in PD if the corresponding vitamin D measure changes by one unit when potential confounders are held constant. Diagnostics to ensure proper model fit, such as residual scatterplots, Quantile–Quantile residual plots, identifying potential outliers based on DFBETAS, and sensitivity analysis with and without the outliers, were conducted prior to final data interpretation.

To determine specific sOTUs associated with each of the five vitamin D metabolite measures, we used random forest classification with 5-fold-within-5-fold nested cross-validation in Python 3.6.10[53]. We first applied random forest to the samples in the top and bottom deciles of each vitamin D measure for classification, and then retained only the models that have high classification accuracy from nested cross-validation, whose average area under the curve (AUC) from 5-fold-within-5-fold cross-validation were >0.7. Nested cross-validation is preferred over the commonly used flat cross-validation, because it gives an unbiased estimate of model's generalization performance[53,54]. In each iteration of the outer cross-validation, the hyperparameters of the random forest model are tuned independently to minimize an inner cross-validation estimate of generalization performance. Hence, the model's performance is essentially estimated by the outer cross-validation. This eliminates the bias introduced by the flat cross-validation procedure, because the test data in each iteration of the outer cross-validation have not been used to optimize the model's performance. Using this criterion of average AUC >0.7 in the nested CV, only random forest models on 1,25(OH)$_2$D and ratio of activation were kept. Using these two models, we then proceeded to define the sOTUs with random forest feature importance scores higher than 0.2% to be potential candidates associated with the vitamin D measure of interest. The importance of a feature in our random forest model (implemented with *sklearn.ensemble.RandomForestClassifier*) is an impurity-based feature importance, computed as the normalized total reduction of the Gini criterion brought by that feature, also known as the Gini importance. Hence, the higher the feature importance, the more the feature contributes to the model fit. To further examine the direction of associations between the sOTUs identified by random forest and vitamin D metabolites, Spearman rank correlation coefficients were used with the significance of correlation determined by Spearman's p-value with BH-FDR correction. Only those sOTUs with Spearman's p-value >0.05 after BH-FDR correction were defined to be significantly associated with the vitamin D measure of interest.

**Reporting summary**. Further information on research design is available in the Nature Research Reporting Summary linked to this article.

## Data availability

The Knight Lab received the microbial sequencing data in the form of FASTQ files from the San Francisco Coordinating Center. The sequences and biom datasets were submitted to the European Bioinformatics Institute (EBI) database in April 2018, with accession number ERP107984 and are publicly available. All other data, including clinical data, are available from the corresponding author upon reasonable request (email: dkado@ucsd. edu). The raw metabolomic data for the clinical measures of vitamin D metabolites are not available as they were performed in a large volume clinical laboratory in Leuven, Belgium where the mass spectrometry data are not routinely kept. However, the source vitamin D data files and all other clinical data are stored in the MrOS data online repository. Participant-level personally identifiable data are protected under the Health Insurance Portability and Accountability Act of 1996 (HIPAA), Public Law 104-1919 that mandated the adoption of Federal protections for individually identifiable health information. Thus public distribution is not allowable, but all study data can be made available as a Limited Data Set through accessing https://mrosonline.ucsf.edu. Interested users can create an account by registering online and signing a Data Use Agreement

(DUA). The DUA stipulates that the data recipient agrees not to Use or Disclose the Limited Data Set for any purpose other than Permitted Uses and Disclosures or as Required by Law. The full DUA is available here: https://mrosonline.ucsf.edu/Account/UserAgreement. Source data are provided with this paper.

## Code availability

All analyses can be found under https://github.com/knightlab-analyses/vitamin-d and https://doi.org/10.5281/zenodo.412357655.

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

## Acknowledgements

We thank Loki Natarajan, PhD for suggestions on statistical analysis. We are also grateful to the MrOS study participants for their continued dedication to this study and to the other principal investigators and staff of the MrOS study centers: (1) James Shikany, University of Alabama, Birmingham, AL, USA; (2) Kristine Ensrud, University of Minnesota, Minneapolis, MN, USA; (3) Jane Cauley, University of Pittsburgh, Pittsburgh, PA, USA; and (4) Marcia Stefanick, Stanford University, Palo Alto, CA, USA. This work was supported by the National Institutes of Health (grant numbers U01 AG027810, U01 AG042124, U01 AG042139, U01 AG042140, U01 AG042143, U01 AG042145, U01 AG042168, U01 AR066160, and UL1 TR000128). We would also like to thankfully acknowledge the Dr. Ruth Covell Fund for Geriatric Research and Education for covering the publication costs.

## Author contributions

R.L.T., E.O., and D.M.K. designed and proposed the research project. L.J., R.L.T., Z.Z.X., J.S.A., E.O., R.K., and D.M.K. planned the experimental work. D.V. and S.P. performed the mass spectrometry experiments. E.O., G.A., L.J., Z.Z.X., S.J., and R.K. carried out microbiome processing and sequencing. L.J., J.S., Z.Z.X., S.J., and R.K. conducted the microbiome classification and data analysis. R.L.T., L.J., and D.M.K. drafted the original manuscript. All authors reviewed, edited, and approved the final draft.

## Competing interests

The authors declare no competing interests.
