## [Peer Review File · Nature Communications]

Reviewers' comments:

Reviewer #1 (Remarks to the Author):

This cross-sectional study was aimed at investigating the relationship between gut microbiome and the host vitamin D signaling in a sample of elderly men from the MrOS Study. Robust correlations between the active vitamin D metabolites, 1,25(OH)₂D and 24,25(OH)₂D, and the gut microbiome were described. In particular, individuals with higher 1,25(OH)₂D levels showed a greater alpha- and beta-diversity, likely reflecting a healthier state. Higher 1,25(OH)₂D levels were associated with a more favorable gut microbial diversity, including specific microbiota that are known butyrate producers. Overall the study is original, well written and raises the following points:

- 1) It could be of interest to include some information about alpha- and beta-diversity in subjects taking antibiotics.
- 2) It is known that vitamin D levels differ in relation to the latitude. Was there any variation concerning vitamin D metabolites and vitamin D metabolic flux in relation to the site?
- 3) If available, PTH levels should be also considered in statistical analysis, given the major role of PTH on vitamin D metabolism and 1-alpha hydroxylation of vitamin D.

Reviewer #2 (Remarks to the Author):

Thomas et al study the relation of vitamin D in the blood circulation and the gut microbiome in men. To this end, they conducted a cross-sectional analysis of 567 elderly men enrolled in the Osteoporotic Fractures in Men Study (MrOS). This is a well-established epidemiological study designed to understand how osteoporosis is related to prostate disease. However, at baseline, participants completed questionnaires regarding medical history in general, medications, physical activity, diet, alcohol intake, and cigarette smoking. The study includes a wide range of objective measures of anthropometric, neuromuscular, vision, strength, and cognitive variables were obtained. Also, serum, urine, and DNA specimens were collected, allowing to address a wide range of clinically relevant parameters related to diabetes, dyslipidemia and inflammation.

I have three major comments.

- In the abstract the authors notice the association is bidirectional. This is not followed through consistently in paper. The introduction states the 'the gut alters intestinal vitamin D' while page 5 states that '1,25(OH)2D was the factor that explained the highest proportion of alpha diversity just over 5%', assuming the vitamin D determines gut microbiome diversity. These contradictions carry through the results and discussion. This is confusing to the reader and has to be harmonized throughout the paper: this study cannot determine the direction.
- Serum vitamin D has been been 'a major hype' in epidemiological research and has been implicated in a wide range of disorders. Reading the paper one is curious how the findings translate into health is imbedded in this rich epidemiological cohort. The impact of the journal as well as the relevance of the finding begs for a series of analysis addressing the question: gut microbiome ↔ vitamin D ↔ health/disease.
- In line with this: Proton pump inhibitors (PPIs) are known to affect the microbiome – why was this medication not included in the analyses? Can the authors check the association?

Other questions

- The authors find clarify the study site effect –which sites were different and why?
- Conducting a random-forest analyses on a relatively small study sample carries the risk of overfitting and false positive findings as for an independent replication study. Alternatively, the authors can consider a regression analysis instead and heat plot?
- To what extend are findings comparable to the mice experiments – can you please integrate the findings in the introduction and discussion in terms of the microbiota associated to vitamin D. Did the animals studies also point to butyrogenic bacteria?
- At page 8 there is a long explanation of vitamin D metabolism – the very general readership of the Nature Com needs to be informed about the many disorders associated to blood vitamin D levels and which of these were bench-marked in prospective epidemiological or by genomic Mendelian Randomization.
- I am confused by the interpretation of the germfree animal experiment (page 10; top): why do the authors exclude that finding that within 2 weeks after convertionalization vitamn D levels go up, implies that the microbiome determines the vitamin D levels rather than vice versa?
- The study would benefit from an association to a health/disease outcome related to vitamin D – for instance osteoporosis/BMD. However, page 11 suggests the authros should also study diabetes. Was there any association to the various diseases studied?
- An association to butyrogenic bacteria is found: is there link between butyrate and vitamin D? Or osteoporosis or any other disease? The fact that the combination of 1,25(OH2D) and butyrate increases the VDR expression in colonic epithelial cells is interesting. Do the authors hypothesize

that this results in higher blood levels ? If yes, this should be stated but raises the question: is there any evidence?

- I cannot follow the discussion in the first paragraph of page 11. Please elucidate what is the point or lesson – if there is any. The text is confusing to a general readership and as it reads now the findings are fully inconsistent? But the jumps from butyrate to glucose and the relevance of vitamin D cannot be followed in the current text.

Reviewer #3 (Remarks to the Author):

The manuscript describes the correlations between the bacterial community and vitamin D metabolites

in faeces of elderly men. Bacterial communities were defined using the 16S rRNA gene. The 16S rRNA

gene does not bring new insights on functions, that should be an important requirement for this kind of study. Moreover, the paper lacks important details about preprocessing of raw data, statistical and machine learning analysis.

Introductory Paragraph

* "16S ribosomal RNA" -> 16S ribosomal RNA gene

* "Random forest analyses identified ... positively associated": how do you define the "positive association"? A Random Forest model is strongly non linear.

Vitamin D metabolites and alpha-diversity

* A recap on redundancy analysis should be added.

* 1,25(OH)₂D was the factor that explained the highest proportion of the variance in alpha-diversity ... at just over 5%: This value seems very low, what is its significance?

* "In multiple linear regression analyses adjusted for ...": which type of multiple linear regression?

Did you standardize/rescale the measurements? If yes, why? How did you manage missing values (if any)?

Please specify. What do you mean when you say "adjusted for"? Please specify.

* p-values should be corrected for multiple testing as for beta diversity analysis.

Vitamin D metabolites and beta-diversity:

* "In redundancy analyses using either unweighted or weighted UniFrac": By looking at the code, it seems that you used the first 10 Principal Coordinates as variables? Why?

* "1,25(OH)₂D explained the highest proportion of variation in microbial beta-diversity (~2%)": This value seems very low, what is its significance?

* Q -> q

* Please report PCoA plots.

* Results regarding non vitamin-D-related factors should be moved in an additional section.

* "not significantly affect subjects' distribution in beta-diversity analysis": please rephrase

Vitamin D metabolites and specific taxonomies

* "12 unique sOTUs were identified to correlate": how? Are you talking about RF feature importance?

* "Six sOTUs were associated with...": how? Are you talking about RF feature importance?

* "Four sOTUs were associated with 24,25(OH)2D levels": how? Are you talking about RF feature importance?

* "Overall, the Firmicutes phylum (for all but two sOTUs), was positively associated": how did you determine

the type of association?

Methods

* Important details about 16S rRNA gene preprocessing are missing.

* Important details about statistical and machine learning analysis are missing (e.g. did you performed cross validation? How? What is the generalization error? ...)

Response to Reviewer's Comments:

Reviewer #1 (Remarks to the Author, *in italics*):

This cross-sectional study was aimed at investigating the relationship between gut microbiome and the host vitamin D signaling in a sample of elderly men from the MrOS Study. Robust correlations between the active vitamin D metabolites, 1,25(OH)₂D and 24,25(OH)₂D, and the gut microbiome were described. In particular, individuals with higher 1,25(OH)₂D levels showed a greater alpha- and beta-diversity, likely reflecting a healthier state. Higher 1,25(OH)₂D levels were associated with a more favorable gut microbial diversity, including specific microbiota that are known butyrate producers. Overall the study is original, well written and raises the following points:

1) It could be of interest to include some information about alpha- and beta-diversity in subjects taking antibiotics.

We agree. Although only 6.7% of the study cohort reported recent antibiotic use in the past 30 days, there were significant effects with reduced alpha diversity as well as altered beta diversity among these men (see Figure 4, reproduced below). Thus, antibiotic use has since been included as a covariate in all reported multivariable analyses, though the overall main findings demonstrating a significant association between active 1,25(OH)₂D and gut microbial alpha and beta diversity remain unchanged. Our data findings are consistent with well-documented pervasive reduction in microbial diversity after antibiotic use (PMCID: [PMC5032909](https://pubmed.ncbi.nlm.nih.gov/290909/)).¹ We have since added additional relevant text to the results (see page 6, lines 15-18).

Figure 4: A. Reduced alpha diversity in patients who have taken antibiotics within in the past 30 days. **B.** Unweighted UniFrac PCoA plot showing significant difference in β-diversity associated with antibiotic use in the past 30 days.

2) It is known that vitamin D levels differ in relation to latitude. Was there any variation considering vitamin D metabolites and vitamin D metabolic flux in relation to site?

We concur that the relationship between vitamin D metabolites, latitude, and associated sun exposure is worth exploring and thank the reviewer for this suggestion. We have added an

additional figure that demonstrates the relationship between study site and each of the vitamin D metabolites (new **Figure 1**). As was reported in our paper, site was a significant covariate for both alpha and beta diversity, but we now also demonstrate that 25(OH)D levels were significantly higher in San Diego compared with other sites, except for Palo Alto that has a similar climate (**Figure 1b**). Considering the number of average annual sunny days, San Diego (263) and Palo Alto (260) share similar sun exposure, while Birmingham of southern latitude, has about 210 days on average* (**Figure 1a**). Interestingly, there was no difference in 1,25(OH)₂D levels between any of the sites (**Figure 1c**). Taken together, these additional data suggest that sun exposure affects storage levels of vitamin 25(OH)D, but not levels of the active vitamin D metabolite 1,25(OH)₂D. It should be noted that the relationship between 25(OH)D levels and MrOS study site was previously reported by Orwoll, et. al. in 2009 (PMCID: [PMC2682464](https://pubmed.ncbi.nlm.nih.gov/2682464/)).² This paper also reported slightly higher total vitamin D levels in San Diego and Palo Alto, with higher values in the summer months and lower values during the winter. It is encouraging that trends from vitamin D quantification in 2009 are reproducible in our study population studied years later using independent mass spectrum analysis. Our study also adds novel analysis of the other vitamin D metabolites. We have updated our revised manuscript to include these observations (see page 5, lines 21-23 and page 6, lines 1-4 and Figure 1a, 1b, and 1c).

Figure 1.

*Data through 2018, demonstrate the number of clear or partially cloudy days (<https://www.ncdc.noaa.gov/gHCN/comparative-climatic-data>)

*Data through 2018, demonstrate the number of clear or partially cloudy days (<https://www.ncdc.noaa.gov/ghcn/comparative-climatic-data>) for San Diego (263), San Francisco as a surrogate for Palo Alto (260), Portland (142), Birmingham (210), Minneapolis (196), and Pittsburgh (162).

3) If available, PTH levels should be also considered in statistical analysis, given the major role of PTH on vitamin D metabolism and 1-alpha hydroxylation of vitamin D.

Unfortunately, we do not have simultaneous serum PTH data available as PTH plays an important role in calcium and vitamin D regulation. That being stated, the distal colon does not express the PTH receptor and thus, local effects of the microbiome on vitamin D metabolism or vice versa independent of PTH might be reasonable to assume.

However, to provide a surrogate marker of PTH, we inferred that in the setting of vitamin D deficiency, the activation ratio should be increased (being physiologically associated with elevated PTH, or secondary hyperparathyroidism). In our study, we compared the activation ratio in subjects with adequate vitamin D versus deficient subjects ($25(\text{OH})\text{D} < 20 \text{ ng/mL}$, $n = 40$), and found that the activation ratio was significantly higher in vitamin D deficient participants ($p = 3.9 \times 10^{-7}$), suggesting that the activation ratio may serve as a surrogate marker for elevated PTH in this group (see **Figure 6a**). Accordingly, in patients who have adequate levels of vitamin D, it would follow logically that they might have higher levels of catabolism, and this is what we found (see **Figure 6b**). We have since added these figures and additional discussion regarding vitamin D metabolism endocrine feedback (see page 10, lines 4-7).

Figure 6.

Reviewer #2 (Remarks to the Author):

Thomas et al study the relation of vitamin D in the blood circulation and the gut microbiome in men. To this end, they conducted a cross-sectional analysis of 567 elderly men enrolled in the Osteoporotic Fractures in Men Study (MrOS). This is a well-established epidemiological study designed to understand how osteoporosis is related to prostate disease. However, at baseline,

participants completed questionnaires regarding medical history in general, medications, physical activity, diet, alcohol intake, and cigarette smoking. The study includes a wide range of objective measures of anthropometric, neuromuscular, vision, strength, and cognitive variables were obtained. Also, serum, urine, and DNA specimens were collected, allowing to address a wide range of clinically relevant parameters related to diabetes, dyslipidemia and inflammation.

I have three major comments.

- In the abstract the authors notice the association is bidirectional. This is not followed through consistently in paper. The introduction states ‘the gut alters intestinal vitamin D’ while page 5 states that ‘1,25(OH)2D was the factor that explained the highest proportion of alpha diversity just over 5%’, assuming the vitamin D determines gut microbiome diversity. These contradictions carry through the results and discussion. This is confusing to the reader and has to be harmonized throughout the paper: this study cannot determine the direction.*

This is an important point. We thank the reviewer for pointing out the inconsistencies in our wording, realizing that although we believe the found association to be bidirectional, it is confusing to the reader; if in certain instances we infer that the gut alters vitamin D while at the same time we discuss how vitamin D metabolites might affect the gut microbiome. In response, we have now corrected these inconsistencies, and since it is convention to determine factors that explain alpha and beta diversity, we have changed the random forest results to read similarly by describing the results as associations between vitamin D metabolites and specific taxonomies rather than vice versa.

- Serum vitamin D has been a ‘major hype’ in epidemiological research and has been implicated in a wide range of disorders. Reading the paper one is curious how the findings translate into health is imbedded in this rich epidemiological cohort. The impact of the journal as well as relevance of the finding begs for a serious of analysis addressing the question: gut microbiome · vitamin D · health/disease.*

We agree with the reviewer that vitamin D and its implications for health have been widely studied and that MrOS is a rich epidemiological cohort that lends itself to studying a multitude of disorders, including its primary initial aims that were to identify fracture risk factors in older men. As this is a multi-site study that includes a consortium of many investigators, there are others who have already studied 25(OH)D levels and published their findings (PMCID: PMC2682464).² Thus, it is not warranted to repeat these same studies, but we have now referenced them in our revised manuscript to provide the reader with additional background on vitamin D and sleep, incident diabetes and cardiovascular events that is specific to the MrOS cohort (PMCID: PMC4288606, PMC5466880 and PMC4154079).³⁻⁵ With regard to expanding the study to examine the associations between the gut microbiome and health/disease, other MrOS investigators have already published or are currently working on a multitude of different gut microbiome projects that address these outcomes (e.g. osteoporosis, aging, obesity, sleep, diabetes, cardiovascular disease, mortality, etc), so it is not feasible to add these health conditions to our current study due to competing overlap.

Previous studies, including the published MrOS studies on vitamin D, 25(OH)D, were only associated with poor sleep, with no significant associations found with incident diabetes or cardiovascular events. In addition to others, these results point to the fact that although 25(OH)D is a good measure of bodily vitamin D stores, it is not a consistent predictor of important health outcomes, especially when vitamin D deficient and adequate subjects are not considered as distinct subgroups. This point is abundantly clear in light of recently published

randomized controlled trial data of vitamin D supplementation reveal no improvement in cardiovascular, cancer, or bone health. (PMCID: PMC6425757, PMID: 31923341).^{6,7} Thus, our study findings of serum levels of the biologically active 1,25(OH)₂D and its related markers of metabolism, showing a bidirectional significant association with the gut microbiome is both novel and compelling.

To address this reviewer concerns, we have since added additional text to the introductory paragraph that provides better context of vitamin D in association with various health outcomes in human populations in addition to citing the most recent vitamin D supplementation randomized controlled trial data that studied over 25,000 older adults (see page 4, lines 3-13 and page 5 lines 2-4).

• *In line with this: Proton pump inhibitors (PPIs) are known to affect the microbiome – why was this medication not included in the analyses? Can the authors check the association?*

We thank the reviewer for pointing out our oversight. We have since included PPI use in our revised analyses. In our study cohort, 19% of the cohort reported using PPI's. In analyses of alpha diversity, we found no effect of PPI use (see Figure directly below), but there was a significant effect in beta-diversity ($q = 0.0006$). We have since added PPI use as a covariate in all reported results, including **Table 1**, **Figure 2b**, and in the text (page 6, line 23 and page 7, lines 1-2).

Other questions

• *The authors find clarify the study site effect –which sites were different and why?*

We have addressed this query above in response to reviewer 1 comments with regards to study site, measures of vitamin D and latitude. In case there are other site-related differences we have not accounted for, we have adjusted for site in all of the statistical models.

• Conducting a random-forest analyses on a relatively small study sample carries the risk of overfitting and false positive findings as for an independent replication study. Alternatively, the authors can consider a regression analysis instead and heat plot?

We considered a regression analysis, but given about 25K sOTUs on around 120 samples, regression methods would have limited capacity in identifying differentially abundant sOTUs without inflated type 1 error. Nonetheless, honoring the reviewer recommendation, we have since performed a regression analysis with L1 regularization (LASSO) and found no sOTUs associated with vitamin D measures.

Because random forest analysis has been reported as one of the most effective learning models for analyzing microbiome data with high classification accuracy, we used this approach in our analyses. Random forest analysis has been successfully demonstrated with a variety of 16S rRNA data sets for identification of body habitat, host, and disease states (PMCID: PMC3960509, PMID: 21039646).^{8,9} In our study, with about 60 samples in each classification category, the sample size was considered adequate for these analyses. To avoid overfitting, we used 5-fold cross-validation based on area under the curve (AUC) criteria to tune the hyperparameters. Then, to reduce the risk of false positive findings, we only retained random forest models with high classification accuracy (higher than 0.7 AUC from 5-fold cross-validation) and only the sOTUs that had random forest feature importance scores higher than 0.2%. Lastly, we used Spearman correlation to examine the direction between the random forest identified sOTUs and vitamin D measures; only the sOTUs with Benjamini-Hochberg corrected p-values (<0.05) were considered significant. In response to the reviewer queries, a more detailed explanation of our analytic approach has been edited for clarification (see page 20, lines 7-21).

• To what extent are findings comparable to the mice experiments – can you please integrate the findings in the introduction and discussion in terms of the microbiota associated to vitamin D. Did the animals studies also point to butyrogenic bacteria?

We appreciate the reviewer recommendation. While we have included more background citing human studies in the introduction (based on the previous reviewer comment), we now elaborate more on the mouse data in the discussion. In addition to highlighting a 2018 study by Bora, et. al. where reconstituting the microbiome in germ free mice increased levels of vitamin D metabolites and improved hypocalcemia (PMID: 29599772) (see page 11, lines 4-10),¹⁰ we now clarify that microbial reconstitution included butyrate producing bacteria (see page 11, lines 9-11). In that experiment they used Schaedler's flora (a group of 8 commensals including several butyrate producers) (PMID: 27824342)¹¹ We have also cited additional mice data of VDR and CYP27B1-hydroxylase deficiency knockout mice that have fewer normal Firmicutes (see page 11, lines 22-23 and page 12, lines 1-3).

• At page 8 there is a long explanation of vitamin D metabolism – the very general readership of the Nature Com needs to be informed about the many disorders associated to blood vitamin D levels and which of these were bench-marked in prospective epidemiological or by genomic Mendelian Randomization. Have studies been published about vitamin D metabolism in the literature?

Thank you for this suggestion. We have included more background information regarding blood vitamin D levels, the different metabolites, and their clinical implications. Namely, we now include more background regarding the bodily stores of vitamin D, (25(OH)D, and how when deficient there is a link to rickets and osteoporosis as well as other diseases such as diabetes,

cardiovascular disease, and cancer, to name a few. That being stated, recently published large randomized controlled trial data where over 25,000 men and women were randomized to vitamin D3 2,000 IU daily versus placebo, there was no effect on outcomes including incident cardiovascular disease, cancer or bone health (PMCID: PMC6425757, PMID: 31923341).^{6,7} This latest “gold standard” evidence from the largest randomized clinical trial to date only strengthens the findings of our manuscript. We posit that it is the biologically active form of vitamin D, 1,25(OH)₂D, not 25(OH)D, and its associated measures of activation and catabolism that have major gut health implications. Moreover, the three other large epidemiological studies point to the measures of vitamin D activation and catabolism being superior to the traditional clinical marker of vitamin D status (PMCID: PMC4661572, PMCID: PMC5794222, PMID: 31891001).^{12,13,14} As such, the current manuscript adds to that literature and is novel in that it provides more of a biologically plausible mechanism, relating the vitamin D metabolome with the gut microbiome. We have since added additional text to provide more background and emphasis on why the current study findings are novel (see page 4, lines 3-13, 16-18, page 5, lines 2-4, page 9, lines 4-7, page 10, lines 19-23).

• I am confused by the interpretation of the germfree animal experiment (page 10; top): why do the authors exclude that finding that within 2 weeks after conversionalization, vitamin D levels go up, implies that the microbiome determines the vitamin D levels rather than vice versa?

We presented prior study publications, the sum of which support bidirectionality of the association between the microbiome and serum vitamin D metabolome. In our discussion we did not exclude the findings, but rather presented previous published related work in animal and human models that overall, support bidirectionality of the association.

• The study would benefit from an association to a health/disease outcome related to vitamin D – for instance osteoporosis/BMD. However, page 11 suggests the authors should also study diabetes. Was there any association to the various diseases studied?

We appreciate this inquiry and have provided additional background regarding vitamin D and outcomes of incident diabetes and cardiovascular disease that have been previously published out of the MrOS cohort (PMCIDs: PMC5466880 and PMC4154079).^{4,5} In these publications, 25(OH)D was not associated with incident diabetes or cardiovascular diseases, similar to our findings that it was not associated with the gut microbiome. See page 4, lines 3-13.

• An association to butyrogenic bacteria is found: is there link between butyrate and vitamin D? Or osteoporosis or any other disease? The fact that the combination of 1,25(OH)₂D and butyrate increases the VDR expression in colonic epithelial cells is interesting. Do the authors hypothesize that this results in higher blood levels? If yes, this should be stated but raises the question: is there any evidence?

We agree that the *in vitro* experiments suggesting an interaction between butyrate, 1,25(OH)₂D, and the colonic VDR are intriguing. However, these results are only preliminary. Further mechanistic work would be needed to confirm these findings.

We have since rewritten this part of the discussion to downplay the potential mechanism as exploratory rather than fact. See page 11, lines 13-18 and 20-23, and page 12, lines 1-18.

• I cannot follow the discussion in the first paragraph of page 11. Please elucidate what it the point or lesson – if there is any. The text is confusing to a general readership and as it reads now the findings are fully inconsistent? But the jumps from butyrate to glucose and the relevance of vitamin D cannot be followed in the current text.

We agree with the reviewer's assessment and have since replaced this paragraph as suggested.

Reviewer #3 (Remarks to the Author):

The manuscript describes the correlations between the bacterial community and vitamin D metabolites in faeces of elderly men. Bacterial communities were defined using the 16S rRNA gene. The 16S rRNA gene does not bring new insights on functions, that should be an important requirement for this kind of study.

We respectfully suggest that this study, using the newest insights into taxonomy and then bridging taxonomy with functional roles of bacteria at a high level in a large, well-defined patient cohort, provides new and original insights to the field of vitamin D metabolism and how it relates to the human gut microbiome.

In addition, since the original analyses were done, we have since rerun the entire analyses with the latest versions of QIITA and QIIME pipelines (2020 instead of 2018). Overall, the results have not changed much, with the main study findings remaining significant. Most changes occurred at the level of taxonomic identification down to the genus rather than just the family or order. The revised manuscript has been updated to reflect these changes (see page 6, line 10; page 7, lines 4-19, and page 8, lines 2-15).

Moreover, the paper lacks important details about preprocessing of raw data, statistical and machine learning analysis.

We appreciate the reviewer's comment. In the revised manuscript, we have provided substantially enhanced detail about the pre-processing of raw data and the machine learning analyses. We have further aimed to address each of the technical concerns raised by the reviewer, as discussed below.

Introductory Paragraph

* "16S ribosomal RNA" -> 16S ribosomal RNA gene

Corrected in the text (page 3, line 6).

* "Random forest analyses identified ... positively associated": how do you define the "positive association"? A Random Forest model is strongly non linear.

The reviewer correctly points out that random forest analyses cannot indicate the direction of associations between identified sOTUs and vitamin D metabolites. In our study, after running the random forest models, we used Spearman correlations to infer the direction of associations. We have since clarified our approach to defining the directions of associations in the text: "To further examine the direction of associations between the sOTUs identified by random forest and vitamin D metabolite, Spearman rank correlation coefficients were used and the significance of correlation was determined by Spearman's p-value with BH-FDR correction." (See page 20, lines 16-21).

Moreover, the footnote underneath Table 2 regarding random forest analyses has now been changed to read, "Direction of associations between taxa identified by random forest and each vitamin D metabolite determined by Spearman rank correlation coefficients. Significance of correlation decided by Spearman's p-value with BH-FDR correction."

Vitamin D metabolites and alpha-diversity

** A recap on redundancy analysis should be added.*

This been accomplished with the following added text: "Non-redundant covariates were identified using a forward stepwise redundancy analysis (RDA) with the `rda` and `ordiR2step` functions in `vegan` package in R. RDA is a direct gradient analysis technique that summarizes the linear relationships between components of response variables that are "redundant" with (i.e. explained by) a set of explanatory variables. Automatic forward selection procedure was applied to RDA to select a subset of explanatory variables (i.e. non-redundant covariates) with a two-step procedure to prevent a highly inflated type 1 error and an overestimation of the amount of explained variance. First, a global test using all explanatory variables was done prior to forward selection. Second, forward selection was carried out with permutation test (1000 permutations) using two stopping criteria: 1) the usual significance level alpha (prespecified as <0.05) and 2) the adjusted coefficient of multiple determination R^2). This analysis estimates the linear cumulative and independent effect size (based upon adjusted R^2) of each non-redundant covariate on microbiome composition variation. (See page 18, lines 21-23, page 19, lines 1-9).

** 1,25(OH)₂D was the factor that explained the highest proportion of the variance in alpha-diversity.... at just over 5%: This value seems very low, what is its significance?*

Yes, interestingly, 1,25(OH)₂D was the factor among 27 variables considered that explained the highest proportion of alpha-diversity in these men. Despite its seemingly low explained proportion of variance at just over 5%, 1,25(OH)₂D is highly significant, with a p-value of 0.002 based on the permutation test from the automated forward selection procedure. From a biological perspective, a 5% effect size is not low compared to the effect sizes reported in Falony et al, 2016, where a set of 18 covariates the largest explanatory power on microbiome composition explained only 7.63% of community variation, and the top single nonredundant covariate explained about 4% of the variation (PMID: 27126039).¹⁵

** "In multiple linear regression analyses adjusted for ...": which type of multiple linear regression?*

We modeled the relationship between a continuous outcome (alpha diversity) and multiple explanatory variables (vitamin D measures and other covariates of interest) by fitting a linear equation to the observed data, with the parameters of interest estimated by minimizing the sum of squares of the residuals.

Did you standardize/rescale the measurements? If yes, why? How did you manage missing values (if any)? Please specify.

Prior to applying multiple linear regression (MLR), we standardized all the vitamin D measures to have zero mean and unit variance. Because the scales of vitamin D measures vary greatly, we chose to standardize our measurements so that the interpretations of estimated parameters would be easier to compare across different measurements. For example, with standardization,

the slopes for different vitamin D measures from MRL models in Figure 2, are directly comparable.

To deal with any missing values, we did a complete-case analysis, where all cases with missing outcomes were excluded; at most, 54 samples of the 599 were removed in all the five MLR models.

What do you mean when you say “adjusted for”? Please specify.

Considering that MLR allows for estimation of the association between a given explanatory variable (one vitamin D measure) and the outcome (alpha diversity) holding all other variables constant, it provides a way of adjusting for (or accounting for) potentially confounding variables that have been included in the model. Specifically, we first used stepwise backward selection with ANOVA type II test to select the confounding variables that significantly affect alpha diversity (without including vitamin D measures). In this way, we identified age, BMI, race, site, antibiotic use, antidepressant use, physical activity score, season of visit, and total starch intake as potential confounders. Then we added each vitamin D measure of interest to the backward selection model to derive the final reported five MLR models, where the estimated coefficient for each vitamin D measure is interpreted as the average change in alpha diversity if the corresponding vitamin D measure changes by one unit when potential confounders are held constant.

**p-values should be corrected for multiple testing as for beta diversity analysis.*

We agree and apologize for any lack of clarity. Our beta diversity analysis using PERMANOVA tests on unweighted UniFrac distance was corrected for multiple testing with the Benjamini-Hochberg (BH) FDR procedure. We have since rephrased the sentence in the methods to better clarify this point: “...PERMANOVA tests with multiple testing correction of the Benjamini-Hochberg FDR procedure...” See page 19 lines 10-13.

Vitamin D metabolites and beta-diversity:

** "In redundancy analyses using either unweighted or weighted UniFrac": By looking at the code, it seems that you used the first 10 Principal Coordinates as variables? Why?*

Although distance-based redundancy analysis (dbRDA) exists, automatic stepwise model selection procedure that prevents inflated Type I error and overestimation of the amount of explained variance is not applicable to dbRDA since it does not allow identification of non-redundant covariates for effect size estimation. Thus, in order to apply automatic stepwise model selection to traditional RDA for beta diversity analyses, we used the first 10 Principal Coordinates from the Principal Coordinate Analysis (PCoA) on unweighted UniFrac that provided a low dimensional approximation to the original beta diversity distance matrix. To ensure that the usage of only 10 principal coordinates did not bias our results, we also repeated the same analysis by using all the principal coordinates and found similar results.

** "1,25(OH)2D explained the highest proportion of variation in microbial beta-diversity (~2%)": This value seems very low, what is its significance?*

As in our response to a similar question described above about 1,25(OH)₂D and alpha-diversity results, in this case for beta-diversity, the effect size of ~2% was also deemed to be highly significant with a p-value of 0.002, based upon the permutation test from the automatic forward selection procedure. The method used for effect size determination was the same as reported by Falony et al, Science 2016, who reported similar effect sizes for the covariates considered in their analysis. Finally, it is worth noting that 1,25(OH)₂D has the explanatory power in microbiome composition, even higher than geographic site or age.

*Q -> q

We have since changed all capital Q's to small q's as requested.

* Please report PCoA plots.

We now include the PCoA plots of the 4 vitamin D measures that were found to be significantly associated with beta diversity using PERMANOVA tests (**Figure 5**, also shown below).

Figure 5: β -diversity PCoA plots reveal significant clustering according to vitamin D metabolite levels; darker colors (corresponding to higher metabolite values) appear more prominently closer to the origin of the PCoA axes.

*Results regarding non vitamin-D-related factors should be moved in an additional section.

We are not 100% certain we understand the request. If, by “non vitamin-D-related factors” the reviewer means the covariates and how they relate to the gut microbiome, we would refer them to **Figure 2** (RDA barplots).

“not significantly affect subjects’ distribution in beta-diversity analysis:” please rephrase

We have since rephrased the sentence as: “In β -diversity testing results with PERMANOVA after BH-FDR correction, consistent with redundancy analysis results, most non-redundant covariates retained statistical significance with the exceptions of 25(OH)D ($q = 0.32$) and age ($q = 0.058$). We also examined Vitamin D 25(OH)D as a dichotomous variable based on the clinical definition of Vitamin D deficiency (25(OH)D < 20 ng/ml) and it made no difference in the results ($q = 0.503$). Thus, neither treating vitamin D as a continuous nor categorical variable had a significant impact on β -diversity in our study sample. (See page 7, lines 2-9).

Vitamin D metabolites and specific taxonomies

** “12 unique sOTUs were identified to correlate”: how? Are you talking about RF feature importance?*

** “Six sOTUs were associated with...”: how? Are you talking about RF feature importance?*

** “Four sOTUs were associated with 24,25(OH)₂D levels”: how? Are you talking about RF feature importance?*

Yes, to all of the above. The 12, 6 and 4 sOTU associations reported were derived from RF feature importance. To further clarify, we used random forest feature importance scores to identify vitamin D associated sOTUs. We first applied random forest to the samples in the top and bottom deciles of each vitamin D measure for classification, then we retained only the models that have high classification accuracy, whose average area under the curve (AUC) from 5-fold cross-validation are greater than 0.70. Then we retained the sOTUs with random forest feature importance scores higher than 0.2% and defined those with BH adjusted p-values from Spearman correlation tests higher than 0.05 to be significantly associated with the vitamin D measure of interest. We have since modified the text within the manuscript to better describe the approach used to identify specific taxonomies associated with the vitamin D measures. (See page 20, lines 7-21).

** “Overall, the Firmicutes phylum (for all but two sOTUs), was positively associated”: how did you determine the type of association?*

After using random forest analyses to identify the sOTUs that were significantly associated with vitamin D metabolites, we used Spearman correlation to infer the direction of associations. We have now clarified this in the methods section stating, “To further examine the direction of associations between the sOTUs identified by random forest and vitamin D metabolite, Spearman rank correlation coefficients were used and the significance of correlation was determined by Spearman’s p-value with BH-FDR correction.” (See page 20, lines 16-21).

Methods

**Important details about 16S rRNA gene preprocessing are missing. Important details about 16S rRNA gene processing are missing.*

We have now included these details below and depending upon the editor's recommendations, can also include them in the manuscript, supplement or online.

We targeted and amplified the V4 region of the 16S rRNA gene by PCR using barcoded primers (PMID: 27822518).¹⁶ V4 paired-end sequencing (PMID: 27822518)¹⁶ was performed using an Illumina MiSeq (La Jolla, CA) with 2x250 cycles according to manufacturer's protocols. Raw V4 sequence reads were demultiplexed using Illumina's bcl2fastq software version 2.20.0.433. Primers were trimmed via cutadapt 1.18 (doi:<https://doi.org/10.14806/ej.17.1.200>)¹⁷ uploaded to Qiita (PMCID: [PMC6235622](https://pubmed.ncbi.nlm.nih.gov/PMC6235622/))¹⁸ and quality controlled using the defaults. Forward reads were trimmed to the first 150 nucleotides. The primary feature table was generated using Deblur 1.1.0 (PMID: 28289731)¹⁹ and can be found in Qiita (qiita.ucsd.edu) as study 11274 with artifact 57316. Sequences can additionally be found in EBI under accession number ERP107984.

** Important details about statistical and machine learning analysis are missing (e.g. did you performed cross validation? How? What is the generalization error? ...)*

We have now added more details in the methods section regarding statistical and machine learning analysis. We did 5-fold cross-validation using AUC criteria to tune the hyperparameters in random forest models and to address issues of generalization error. We identified optimal hyperparameters from the models with the highest mean AUC from 5-fold cross-validation. The mean AUCs from 5-fold cross-validation were 0.74 for random forest model on ratio of activation, 0.67 for ratio of catabolism, 0.75 for 1,25(OH)₂D, 0.65 for 24,25(OH)₂D and 0.63 for 25(OH)D. We then retained models with greater than 0.70 AUC for further sOTUs identification.

References (in case PMID/PMCID links do not work):

1. Yassour M, Vatanen T, Siljander H, et al. Natural history of the infant gut microbiome and impact of antibiotic treatment on bacterial strain diversity and stability. *Science translational medicine* 2016;8:343ra81.
2. Orwoll E, Nielson CM, Marshall LM, et al. Vitamin D deficiency in older men. *The Journal of clinical endocrinology and metabolism* 2009;94:1214-22.
3. Massa J, Stone KL, Wei EK, et al. Vitamin D and actigraphic sleep outcomes in older community-dwelling men: the MrOS sleep study. *Sleep* 2015;38:251-7.
4. Napoli N, Schafer AL, Lui LY, et al. Serum 25-hydroxyvitamin D level and incident type 2 diabetes in older men, the Osteoporotic Fractures in Men (MrOS) study. *Bone* 2016;90:181-4.
5. Bajaj A, Stone KL, Peters K, et al. Circulating vitamin D, supplement use, and cardiovascular disease risk: the MrOS Sleep Study. *J Clin Endocrinol Metab* 2014;99:3256-62.
6. Manson JE, Cook NR, Lee IM, et al. Vitamin D Supplements and Prevention of Cancer and Cardiovascular Disease. *The New England journal of medicine* 2019;380:33-44.
7. LeBoff MS, Chou SH, Murata EM, et al. Effects of Supplemental Vitamin D on Bone Health Outcomes in Women and Men in the VITamin D and Omega-3 Trial (VITAL). *Journal of bone and mineral research : the official journal of the American Society for Bone and Mineral Research* 2020.
8. Statnikov A, Henaff M, Narendra V, et al. A comprehensive evaluation of multiclassification methods for microbiomic data. *Microbiome* 2013;1:11.
9. Knights D, Costello EK, Knight R. Supervised classification of human microbiota. *FEMS microbiology reviews* 2011;35:343-59.

10. Bora SA, Kennett MJ, Smith PB, Patterson AD, Cantorna MT. The Gut Microbiota Regulates Endocrine Vitamin D Metabolism through Fibroblast Growth Factor 23. *Frontiers in immunology* 2018;9:408.
11. Biggs MB, Medlock GL, Moutinho TJ, et al. Systems-level metabolism of the altered Schaedler flora, a complete gut microbiota. *Isme j* 2017;11:426-38.
12. Pasquali M, Tartaglione L, Rotondi S, et al. Calcitriol/calcifediol ratio: An indicator of vitamin D hydroxylation efficiency? *BBA Clin* 2015;3:251-6.
13. Ginsberg C, Katz R, de Boer IH, et al. The 24,25 to 25-hydroxyvitamin D ratio and fracture risk in older adults: The cardiovascular health study. *Bone* 2018;107:124-30.
14. Bansal N, Katz R, Appel L, et al. Vitamin D Metabolic Ratio and Risks of Death and CKD Progression. *Kidney international reports* 2019;4:1598-607.
15. Falony G, Joossens M, Vieira-Silva S, et al. Population-level analysis of gut microbiome variation. *Science* 2016;352:560-4.
16. Walters W, Hyde ER, Berg-Lyons D, et al. Improved Bacterial 16S rRNA Gene (V4 and V4-5) and Fungal Internal Transcribed Spacer Marker Gene Primers for Microbial Community Surveys. *mSystems* 2016;1.
17. Martin M. Cutadapt removes adapter sequences from high-throughput sequencing reads. *2011* 2011;17:3.
18. Gonzalez A, Navas-Molina JA, Kosciolk T, et al. Qiita: rapid, web-enabled microbiome meta-analysis. *Nat Methods* 2018;15:796-8.
19. Amir A, McDonald D, Navas-Molina JA, et al. Deblur Rapidly Resolves Single-Nucleotide Community Sequence Patterns. *mSystems* 2017;2.

REVIEWER COMMENTS

Reviewer #1 (Remarks to the Author):

All the points raised by this reviewer have been adequately addressed.

Reviewer #3 (Remarks to the Author):

I appreciate the efforts that the authors have made in response to my concerns about the machine learning models. However, the machine learning procedures implemented in this work lead to have biased results.

When you use cross validation for parameter tuning, the samples used for validation become part of your model. So you need another independent samples to correctly measure the model's performance (e.g. generalization error), performing a nested CV:

- inner CV loop -> grid search
- outer CV loop -> unbiased generalization performance estimation

Moreover, the optimal number of features should be chosen within the grid search procedure (inner CV) and not using arbitrary thresholds like 0.01 or 0.002 on the optimal model (see the selection bias problem).

Other issues:

- Please explain in the paper how the feature importance is computed in sklearn Random Forest models.
- Figure 5, caption: "appear more prominently closer to the origin of the PCoA axes" -> how can you say that?

- Redundancy analyses: please report in the paper that you used the first 10 Principal Coordinates AND show

that by using all the principal coordinates you find similar results.

Reviewer #4 (Remarks to the Author):

Given the current interest in vitamin D, this is a potentially interesting study although the effect size is reasonably small and I was surprised that no attempt was made to revisit some of the medical results of the study given this new association.

I have only a minor comment. Some of the mass spectrometry methods were a little confusing to follow, especially as you refer to 4 separate papers, some of which are behind paywalls. Line 383 appears to suggest a smaller transition for the d6 labelled compound than for its partner label-free compound. Is this a mistake? Line 385 mentions lithium adducts for example, but no lithium is reported as being added.

No quality control data was given for these experiments or the measured instrumental technical error. How many batches were these samples run in and were there batch to batch differences in the quantification? How were these accounted for in processing the results?

RESPONSE TO REVIEWER COMMENTS:

Reviewer #1 (Remarks to the Author, *in italics*):

All the points raised by this reviewer have been adequately addressed.

Thank you.

Reviewer #3 (Remarks to the Author):

I appreciate the efforts that the authors have made in response to my concerns about the machine learning models. However, the machine learning procedures implemented in this work lead to have biased results.

When you use cross validation for parameter tuning, the samples used for validation become part of your model. So you need another independent samples to correctly measure the model's performance (e.g. generalization error), performing a nested CV:

- inner CV loop -> grid search*
- outer CV loop -> unbiased generalization performance estimation*

We thank the reviewer for pointing out the potential optimistic bias in the estimation of our models' performance, which could be caused by the re-use of data in both performance evaluation and hyperparameter tuning. As the reviewer recommended, we have performed a 5-fold-within-5-fold nested cross-validation procedure to re-evaluate our model's generalization performance. Specifically, we performed hyperparameter selection for random forest in the inner 5-fold cross-validation, and then estimate the model's generalization performance in the outer 5-fold cross-validation. Compared to our previous model performance estimation given by flat cross-validation, nested CV gives mostly similar estimates of generalization performance in terms of the average AUC values in 5 outer folds. For our 5 random forest models using each vitamin D measurements as the response and the microbiome data as the features, flat CV reports an average AUC of 0.75 for 1,25(OH)₂D, 0.74 for Ratio of Activation, 0.67 for Ratio of Catabolism, 0.65 for 24,25(OH)₂D and 0.63 for 25(OH)D, and nested CV estimates an average AUC of 0.73 for 1,25(OH)₂D, 0.77 for ratio of activation, 0.62 for ratio of catabolism, 0.65 for 24,25(OH)₂D and 0.52 for 25(OH)D. On the one hand, there was relatively high optimistic bias (AUC difference between nested and flat CV ≥ 0.05) for the models on ratio of catabolism and 25(OH)D; however, these two models weren't considered for feature selection due to their low AUCs in our previous analysis. On the other hand, the two vitamin D measurements 1,25(OH)₂D and ratio of activation, on which we performed future feature selections, received similarly high estimates of model generalization performance (AUC > 0.7) from both nested and flat CVs. Although our conclusions of high prediction accuracies in models for 1,25(OH)₂D and ratio of activation do not change, we appreciate the reviewer's recommendation of nested CV as a better approach to guard against overly optimistic model performance estimation, and did observe this in 2 out of our 5 models. Hence, we have revised our manuscript to use nested CV instead of flat CV for model performance evaluation as below:

"We first applied random forests classification to the samples in the top and bottom deciles of each vitamin D measure for classification, then retained only the models that have high classification accuracy from nested cross-validation, whose average area under the curve (AUC) from 5-fold-within-5-fold cross-validation were greater than 0.7. Nested cross-validation is

preferred over the commonly used flat cross-validation, because it gives an unbiased estimate of model's generalization performance (Wainer and Cawley 2018, Cawley and Talbot 2010). In each iteration of the outer cross-validation, the hyperparameters of the random forest model are tuned independently to minimize an inner cross-validation estimate of generalization performance. Hence, the model's performance is essentially estimated by the outer cross-validation. This eliminates the bias introduced by the flat cross-validation procedure, because the test data in each iteration of the outer cross-validation have not been used to optimize the model's performance. Using this criterion of average AUC greater than 0.7 in the nested CV, only random forest models on 1,25(OH)₂D and ratio of activation were kept.”(see page 22, lines 20-23, and page 23, lines 1-11).

Moreover, the optimal number of features should be chosen within the grid search procedure (inner CV) and not using arbitrary thresholds like 0.01 or 0.002 on the optimal model (see the selection bias problem).

We understand the reviewer's concern about selection bias in our feature selection procedure. As the reviewer suggested, the selection bias could be corrected by performing a nested CV, where the features are selected within the inner cross-validation, and the model performance is evaluated on the outer cross-validation. However, because our dataset contains >25k features, automated feature selection methods, such as recursive feature elimination, coupled with nested cross-validation, are computationally infeasible and might also yield zero selected features, because the intersection of common features selected by 5 inner cross-validations might not exist due to our large number of features. We investigated the original concern about selection bias described by Ambroise and McLachlan (Ambroise and McLachlan 2002), and found that our feature selection procedure is different from the problematic approaches they described. Ambroise and McLachlan's concern is that during the feature selection process, all the available samples were used to carry out the feature selection, and then the same data is used for random partitioning of training and test sets in order to evaluate the model accuracy. In this case, selection bias is introduced because the test error is based on a subset of data used for feature selection. However, our feature selection step is separated from model evaluation, and we did not use the same data for both feature selection and model prediction. Instead, we first find reliable models by requiring them to have at least 0.7 average AUC from cross-validation (the two selected random forests models on 1,25(OH)₂D and ratio of activation were found to have consistent estimates of prediction accuracies from both flat and nested CVs). Then we start feature selection with no additional model selection, by investigating features with high feature importance scores from random forest, and further avoiding false positives with an additional step of Spearman's rank correlation test with BH-FDR correction for multiple testing. We agree that our choice of threshold of 0.2% on feature importance score is subjective, but this step only serves the purpose of reducing the amount of features for further hypothesis testing, and our final sets of features are decided objectively by the Spearman test with BH-FDR correction. Hence, the false positive rates are controlled in our selected features, although we agree that our subjective feature importance threshold might have filtered out some meaningful features before they can be tested. Because our main goal is to guard against false positives rather than false negatives, we believe that our current feature selection approach is appropriate for our analysis.

Other issues:

- Please explain in the paper how the feature importance is computed in sklearn Random Forest models.

We thank the reviewer for pointing this issue out, and have added the explanation of feature importance in our method section as below:

“The importance of a feature in our random forest model (implemented with *sklearn.ensemble.RandomForestClassifier*) is an impurity-based feature importance, computed as the normalized total reduction of the Gini criterion brought by that feature, also known as the Gini importance. Hence, the higher the feature importance, the more the feature contributes to the model fit.” (see page 23, lines 14-18).

- *Figure 5, caption: "appear more prominently closer to the origin of the PCoA axes" -> how can you say that?*

We agreed that this statement is subjective and lacks objective measurements, so we have revised the caption of figure 5 as follows:

Unweighted UniFrac β -diversity PCoA plots of vitamin D metabolites. β -diversity is significantly stratified according to 1,25(OH)₂D, 24,25(OH)₂D, vitamin D activation, and vitamin D catabolism ratios based on PERMANOVA test after BH-FDR correction. Darker colors correspond to higher metabolite values.

- *Redundancy analyses: please report in the paper that you used the first 10 Principal Coordinates AND show that by using all the principal coordinates you find similar results.*

We thank the reviewer for pointing this out. We have added the details of confirming our RDA results of the first 10 principal coordinates by using all the principal coordinates in the method section, as below:

When applying RDA to β -diversity, because distance-based redundancy analysis with automatic stepwise model selection procedure that prevents inflated type I error is not available, we used the first 10 Principal Coordinates from PCoA on unweighted UniFrac that provided a low dimensional approximation to the original β -diversity distance matrix. To ensure that the usage of only 10 principal coordinates did not bias our results, we also repeated the same analysis by using all the principal coordinates. The same set of 11 non-redundant covariates was found to be significant as in the analysis with 10 principal coordinates (See page 21, lines 16-22, page 22, lines 1-2).

Reviewer #4 (Remarks to the Author, *in italics*):

Given the current interest in vitamin D, this is a potentially interesting study although the effect size is reasonably small and I was surprised that no attempt was made to revisit some of the medical results of the study given this new association.

I have only a minor comment. Some of the mass spectrometry methods were a little confusing to follow, especially as you refer to 4 separate papers, some of which are behind paywalls. Line 383 appears to suggest a smaller transition for the d6 labelled compound than for its partner label-free compound. Is this a mistake? Line 385 mentions lithium adducts for example, but no lithium is reported as being added.

No quality control data was given for these experiments or the measured instrumental technical error. How many batches were these samples run in and were there batch to batch differences in the quantification? How were these accounted for in processing the results?

We thank the reviewer for the overall positive review and the useful comments. We believe the modifications in response to these comments will further strengthen and clarify the paper (see page 15, lines 19-23; page 16, lines 1-23, page 17, lines 1-23, page 18, lines 1-2, 12-13, page 19, 9-23, and page 20, lines 1-2). We agree that the mass spectrometry method description for especially 1,25(OH)₂D₃ and 24,25(OH)₂D₃ is confusing and can be difficult to follow. We repeated the key features of this earlier published methodology in this paper and added the cardinal performance characteristics. Lithium is in the buffers (as 0.5 mM lithium acetate in water or methanol) that generate the gradient over the second chromatographic dimension. The transition for the labelled partner of 1,25(OH)₂D is indeed smaller as the reviewer correctly noted. We agree this is not typical. The origin of this difference is that during method development in 2010 by Casetta et al (European Journal of Mass Spectrometry), it was noted that this transition showed less interference in the MS trace for the labelled 1,25(OH)₂D₃. As the same parents are used, most variation effects (e.g. ionization, extraction, injection) are normally fairly compensated. This transition was never changed as the method never showed problems and has acceptable performance (external QC samples (vitamin D External Quality Assessment Scheme, DEQAS), method comparison (Vanderschueren et al. JCEM 2013), internal QC samples).

Due to the large number of samples, the limited time available on the mass spectrometers (these instruments are also used for (other) clinical analyses), and the extended run time, a substantial number of batches were needed. Minor batch-to-batch biases are indeed inevitable, as well as minor variations in extraction, pipetting, injection, and ionization for each individual measurement. The combined effect of all these sources of random biases/variations can be estimated by the between-run imprecision on a repeat serum. Efforts are typically made to minimize and keep these effects between limits by performing a calibration in each run, the use of isotopic internal standards, statistical process control (repeat serum in each run that has to remain between preset limits before accepting batches), and external quality control schemes (DEQAS (4 times 5 samples every year)). In the paper, we added the number of batches (n=7 for 25(OH)D; n=57 for 1,25(OH)₂D and 24,25(OH)₂D), the calibration procedure (in every batch) and quality control procedure (repeat sera in every batch kept between preset limits) as well as the QC data (between-run imprecision on relevant concentration level of repeat serum: 6.7% at 40 pg/mL for 1 α ,25(OH)₂D₃ and 7.6% at 2.0 ng/mL for 24,25(OH)₂D₃; 5.6% at 28.9 ng/mL 25(OH)D; the median concentrations of 1 α ,25(OH)₂D₃, 24,25(OH)₂D₃ and 20(OH)D in the samples of this study were 56 pg/mL, 3.2 ng/mL, and 34.2 ng/mL, respectively) and the external quality assessment scheme (DEQAS) in which the laboratory participates (with acceptable results). Despite all these measures, some spread on the individual results based on minor random calibration, pipetting, extraction, injection, mass spectrometer variations, remains inevitable (as shown in the between-run imprecision). This can indeed lead to more difficulties in finding statistical associations due to 'noise' on the individual measurements. In this study, this is alleviated by the large number of samples spread in multiple batches.

REFERENCES:

- Wainer J, Cawley G. Nested cross-validation when selecting classifiers is overzealous for most practical applications. arXivorg 2018;arXiv:1809.09446.
- Cawley G, Talbot N. On Over-fitting in Model Selection and Subsequent Selection Bias in Performance Evaluation. Journal of Machine Learning Research 2010;11:2079-107.

- Ambroise C, McLachlan G. J. Selection bias in gene extraction on the basis of microarray gene-expression data. *Proceedings of the National Academy of Sciences of the United States of America* 2002; 99(10): 6562-6566.
- Casetta B, Jans I, Billen J, Vanderschueren D, Bouillon R. Development of a method for the quantification of 1 α ,25(OH) $_2$ -vitamin D $_3$ in serum by liquid chromatography tandem mass spectrometry without derivatization. *European journal of mass spectrometry* (Chichester, England) 2010;16:81-9.
- Vanderschueren D, Pye SR, O'Neill TW, et al. Active vitamin D (1,25-dihydroxyvitamin D) and bone health in middle-aged and elderly men: the European Male Aging Study (EMAS). *J Clin Endocrinol Metab* 2013; 95:995–1005.

REVIEWERS' COMMENTS

Reviewer #3 (Remarks to the Author):

All the points raised have been addressed.

Thank you

Reviewer #4 (Remarks to the Author):

The authors have satisfactorily answered my points. I would remark only that given the intra and inter-batch variation that can occur in mass spectrometry, I would recommend they use more QCs per batch in future studies.

The authors thank the reviewer for the questions and requests for clarifications regarding the mass spectrometry vitamin D assays. As this work indicates that more investigation is indicated regarding the vitamin D metabolites and health outcomes, we will be sure to ensure optimal QC procedures given the intra- and inter-batch variation that is inherent with these measures.